# Guarded Policy Optimization with Imperfect Online Demonstrations

**Zhenghai Xue**[1], **Zhenghao Peng**[2], **Quanyi Li**[3], **Zhihan Liu**[4], **Bolei Zhou**[2]
[1]Nanyang Technological University, Singapore, [2] University of California, Los Angeles,
[3]The University of Edinburgh, [4]Northwestern University

## Abstract

The Teacher-Student Framework (TSF) is a reinforcement learning setting where a teacher agent guards the training of a student agent by intervening and providing online demonstrations. Assuming optimal, the teacher policy has the perfect timing and capability to intervene in the learning process of the student agent, providing safety guarantee and exploration guidance. Nevertheless, in many real-world settings it is expensive or even impossible to obtain a well-performing teacher policy. In this work, we relax the assumption of a well-performing teacher and develop a new method that can incorporate arbitrary teacher policies with modest or inferior performance. We instantiate an Off-Policy Reinforcement Learning algorithm, termed **T**eacher-**S**tudent **S**hared **C**ontrol (**TS2C**), which incorporates teacher intervention based on trajectory-based value estimation. Theoretical analysis validates that the proposed TS2C algorithm attains efficient exploration and substantial safety guarantee without being affected by the teacher's own performance. Experiments on various continuous control tasks show that our method can exploit teacher policies at different performance levels while maintaining a low training cost. Moreover, the student policy surpasses the imperfect teacher policy in terms of higher accumulated reward in held-out testing environments. Code is available at `https://metadriverse.github.io/TS2C`.

## 1 Introduction

In Reinforcement Learning (RL), the Teacher-Student Framework (TSF) (Zimmer et al., 2014; Kelly et al., 2019) incorporates well-performing neural controllers or human experts as teacher policies in the learning process of autonomous agents. At each step, the teacher guards the free exploration of the student by intervening when a specific intervention criterion holds. Online data collected from both the teacher policy and the student policy will be saved into the replay buffer and exploited with Imitation Learning or Off-Policy RL algorithms. Such a guarded policy optimization pipeline can either provide safety guarantee (Peng et al., 2021) or facilitate efficient exploration (Torrey & Taylor, 2013).

The majority of RL methods in TSF assume the availability of a well-performing teacher policy (Spencer et al., 2020; Torrey & Taylor, 2013) so that the student can properly learn from the teacher's demonstration about how to act in the environment. The teacher intervention is triggered when the student acts differently from the teacher (Peng et al., 2021) or when the teacher finds the current state worth exploring (Chisari et al., 2021). This is similar to imitation learning where the training outcome is significantly affected by the quality of demonstrations (Kumar et al., 2020; Fujimoto et al., 2019). Thus with current TSF methods if the teacher is incapable of providing high-quality demonstrations, the student will be misguided and its final performance will be upper-bounded by the performance of the teacher. However, it is time-consuming or even impossible to obtain a well-performing teacher in many real-world applications such as object manipulation with robot arms (Yu et al., 2020a) and autonomous driving (Li et al., 2022a). As a result, current TSF methods will behave poorly with a less capable teacher.

In the real world, the coach of Usain Bolt does not necessarily need to run faster than Usain Bolt. Is it possible to develop a new interactive learning scheme where a student can outperform the teacher while retaining safety guarantee from it? In this work we develop a new guarded policy optimization

method called **T**eacher-**S**tudent **S**hared **C**ontrol (**TS2C**). It follows the setting of a teacher policy and a learning student policy, but relaxes the requirement of high-quality demonstrations from the teacher. A new intervention mechanism is designed: Rather than triggering intervention based on the similarity between the actions of teacher and student, the intervention is now determined by a trajectory-based value estimator. The student is allowed to conduct an action that deviates from the teacher's, as long as its expected return is promising. By relaxing the intervention criterion from step-wise action similarity to trajectory-based value estimation, the student has the freedom to act differently when the teacher fails to provide correct demonstration and thus has the potential to outperform the imperfect teacher. We conduct theoretical analysis and show that in previous TSF methods the quality of the online data-collecting policy is upper-bounded by the performance of the teacher policy. In contrast, TS2C is not limited by the imperfect teacher in upper-bound performance, while still retaining a lower-bound performance and safety guarantee.

Experiments on various continuous control environments show that under the newly proposed method, the learning student policy can be optimized efficiently and safely under different levels of teachers while other TSF algorithms are largely bounded by the teacher's performance. Furthermore, the student policies trained under the proposed TS2C substantially outperform all baseline methods in terms of higher efficiency and lower test-time cost, supporting our theoretical analysis.

## 2 BACKGROUND

### 2.1 RELATED WORK

**The Teacher-Student Framework**    The idea of transferring knowledge from a teacher policy to a student policy has been explored in reinforcement learning (Zimmer et al., 2014). It improves the learning efficiency of the student policy by leveraging a pretrained teacher policy, usually by adding auxiliary loss to encourage the student policy to be close to the teacher policy (Schmitt et al., 2018; Traoré et al., 2019). Though our method follows teacher-student transfer framework, an optimal teacher is not a necessity. During training, agents are fully controlled by either the student (Traoré et al., 2019; Schmitt et al., 2018) or the teacher policy (Rusu et al., 2016), while our method follows intervention-based RL where a mixed policy controls the agent. Other attempts to relax the need of well-performing teacher models include student-student transfer (Lin et al., 2017; Lai et al., 2020), in which heterogeneous agents exchange knowledge through mutual regularisation (Zhao & Hospedales, 2021; Peng et al., 2020).

**Learning from Demonstrations**    Another way to exploit the teacher policy is to collect static demonstration data from it. The learning agent will regard the demonstration as optimal transitions to imitate from. If the data is provided without reward signals, agent can learn by imitating the teacher's policy distribution (Ly & Akhloufi, 2020), matching the trajectory distribution (Ho & Ermon, 2016; Xu et al., 2019) or learning a parameterized reward function with inverse reinforcement learning (Abbeel & Ng, 2004; Fu et al., 2017). With additional reward signals, agents can perform Bellman updates pessimistically, as most offline reinforcement learning algorithms do (Levine et al., 2020). The conservative Bellman update can be performed either by restricting the overestimation of Q-function learning (Fujimoto et al., 2019; Kumar et al., 2020) or by involving model-based uncertainty estimation (Yu et al., 2020b; Chen et al., 2021b). In contrast to the offline learning from demonstration, in this work we focus on the online deployment of teacher policies with teacher-student shared control and show its superiority in reducing the state distributional shift, improving efficiency and ensuring training-time safety.

**Intervention-based Reinforcement Learning**    Intervention-based RL enables both the expert and the learning agent to generate online samples in the environment. The switch between policies can be random (Ross et al., 2011), rule-based (Parnichkun et al., 2022) or determined by the expert, either through the manual intervention of human participants (Abel et al., 2017; Chisari et al., 2021; Li et al., 2022b) or by referring to the policy distribution of a parameterized expert (Peng et al., 2021). More delicate switching algorithms include RCMP (da Silva et al., 2020) which asks for expert advice when the learner's action has high estimated uncertainty. RCMP only works for agents with discrete action spaces, while we investigate continuous action space in this paper. Also, Ross & Bagnell (2014) and Sun et al. (2017) query the expert to obtain the optimal value function, which is used to guide the expert intervention. These switching mechanisms assume the expert policy to be optimal, while our proposed algorithm can make use of a suboptimal expert policy. To exploit

samples collected with different policies, Ross et al. (2011) and Kelly et al. (2019) compute behavior cloning loss on samples where the expert policy is in control and discard those generated by the learner. Other algorithms (Mandlekar et al., 2020; Chisari et al., 2021) assign positive labels on expert samples and compute policy gradient loss based on the pseudo reward. Some other research works focus on provable safety guarantee with shared control (Peng et al., 2021; Wagener et al., 2021), while we provide an additional lower-bound guarantee of the accumulated reward for our method.

## 2.2 NOTATIONS

We consider an infinite-horizon Markov decision process (MDP), defined by the tuple $M = \langle \mathcal{S}, \mathcal{A}, P, R, \gamma, d_0 \rangle$ consisting of a finite state space $\mathcal{S}$, a finite action space $\mathcal{A}$, the state transition probability distribution $P : \mathcal{S} \times \mathcal{A} \times \mathcal{S} \to [0, 1]$, the reward function $R : \mathcal{S} \times \mathcal{A} \to [R_{\min}, R_{\max}]$, the discount factor $\gamma \in (0, 1)$ and the initial state distribution $d_0 : \mathcal{S} \to [0, 1]$. Unless otherwise stated, $\pi$ denotes a stochastic policy $\pi : \mathcal{S} \times \mathcal{A} \to [0, 1]$. The state-action value and state value functions of $\pi$ are defined as $Q^\pi(s, a) = \mathbb{E}_{s_0=s, a_0=a, a_t \sim \pi(\cdot|s_t), s_{t+1} \sim p(\cdot|s_t, a_t)} \left[ \sum_{t=0}^\infty \gamma^t R(s_t, a_t) \right]$ and $V^\pi(s) = \mathbb{E}_{a \sim \pi(\cdot|s)} Q^\pi(s, a)$. The optimal policy is expected to maximize the accumulated return $J(\pi) = \mathbb{E}_{s \sim d_0} V^\pi(s)$.

The Teacher-Student Framework (TSF) models the shared control system as the combination of a teacher policy $\pi_t$ which is pretrained and fixed and a student policy $\pi_s$ to be learned. The actual actions applied to the agent are deduced from a mixed policy of $\pi_t$ and $\pi_s$, where $\pi_t$ starts generating actions when intervention happens. The details of the intervention mechanism are described in Sec. 3.2. The goal of TSF is to improve the training efficiency and safety of $\pi_s$ with the involvement of $\pi_t$. The discrepancy between $\pi_t$ and $\pi_s$ on state $s$, termed as policy discrepancy, is the $L_1$-norm of output difference: $\|\pi_t(\cdot|s) - \pi_s(\cdot|s)\|_1 = \int_{\mathcal{A}} |\pi_t(a|s) - \pi_s(a|s)| \, \mathrm{d}a$. We define the discounted state distribution under policy $\pi$ as $d_\pi(s) = (1 - \gamma) \sum_{t=0}^\infty \gamma^t \Pr(s_t = s; \pi, d_0)$, where $\Pr^\pi(s_t = s; \pi, d_0)$ is the state visitation probability. The state distribution discrepancy is defined as the difference in $L_1$-norm of the discounted state distributions deduced from two policies: $\|d_{\pi_t} - d_{\pi_s}\|_1 = \int_{\mathcal{S}} |d_{\pi_t}(s) - d_{\pi_s}(s)| \, \mathrm{d}s$.

## 3 GUARDED POLICY OPTIMIZATION WITH ONLINE DEMONSTRATIONS

Fig. 1 shows an overview of our proposed method. In addition to the conventional single-agent RL setting, we include a teacher policy $\pi_t$ in the training loop. The term "teacher" only indicates that the role of this policy is to help the student training. No assumption on the optimality of the teacher is needed. The teacher policy is first used to do warmup rollouts and train a value estimator. During the training of the student policy, both $\pi_s$ and $\pi_t$ receive current state $s$ from the environment. They propose actions $a_s$ and $a_t$, and then a value-based intervention function $\mathcal{T}(s)$ determines which action should be taken and applied to the environment. The student policy is then updated with data collected through such intervention.

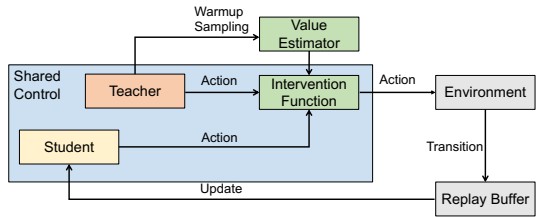

Figure 1: Overview of the proposed teacher-student shared control method. Both student and teacher policies are in the training loop and the shared control occurs based on the intervention function.

We first give a theoretical analysis on the general setting of intervention-based RL in Sec. 3.1. We then discuss the properties of different forms of intervention function $\mathcal{T}$ in Sec. 3.2. Based on these analyses, we propose a new algorithm for teacher-student shared control in Sec. 3.3. All the proofs in this section are included in Appendix A.1.

### 3.1 ANALYSIS ON INTERVENTION-BASED RL

In intervention-based RL, the teacher policy and the student policy act together and become a mixed behavior policy $\pi_b$. The intervention function $\mathcal{T}(s)$ determines which policy is in charge. Let

$\mathcal{T}(s) = 1$ denotes the teacher policy $\pi_t$ takes control and $\mathcal{T}(s) = 0$ means otherwise. Then $\pi_b$ can be represented as $\pi_b(\cdot|s) = \mathcal{T}(s)\pi_t(\cdot|s) + (1 - \mathcal{T}(s))\pi_s(\cdot|s)$.

One issue with the joint control is that the student policy $\pi_s$ is trained with samples collected by the behavior policy $\pi_b$, whose action distribution is not always aligned with $\pi_s$. A large state distribution discrepancy between two policies $\|d_{\pi_b} - d_{\pi_s}\|_1$ can cause distributional shift and ruin the training. A similar problem exists in behavior cloning (BC), though in BC no intervention is involved and $\pi_s$ learns from samples all collected by the teacher policy $\pi_t$. To analyze the state distribution discrepancy in BC, we first introduce a useful lemma (Achiam et al., 2017).

**Lemma 3.1.** *The state distribution discrepancy between the teacher policy $\pi_t$ and the student policy $\pi_s$ is bounded by their expected policy discrepancy:*

$$\|d_{\pi_t} - d_{\pi_s}\|_1 \leqslant \frac{\gamma}{1 - \gamma} \mathbb{E}_{s \sim d_{\pi_t}} \|\pi_t(\cdot \mid s) - \pi_s(\cdot \mid s)\|_1. \tag{1}$$

We apply the lemma to the setting of intervention-based RL and derive a bound for $\|d_{\pi_b} - d_{\pi_s}\|_1$.

**Theorem 3.2.** *For any behavior policy $\pi_b$ deduced by a teacher policy $\pi_t$, a student policy $\pi_s$ and an intervention function $\mathcal{T}(s)$, the state distribution discrepancy between $\pi_b$ and $\pi_s$ is bounded by*

$$\|d_{\pi_b} - d_{\pi_s}\|_1 \leqslant \frac{\beta\gamma}{1 - \gamma} \mathbb{E}_{s \sim d_{\pi_b}} \|\pi_t(\cdot \mid s) - \pi_s(\cdot \mid s)\|_1, \tag{2}$$

*where $\beta = \frac{\mathbb{E}_{s \sim d_{\pi_b}}\left[\mathcal{T}(s)\|\pi_t(\cdot|s) - \pi_s(\cdot|s)\|_1\right]}{\mathbb{E}_{s \sim d_{\pi_b}}\|\pi_t(\cdot|s) - \pi_s(\cdot|s)\|_1} \in [0, 1]$ is the expected intervention rate weighted by the policy discrepancy.*

Both Eq. 1 and Eq. 2 bound the state distribution discrepancy by the difference in per-state policy distributions, but the upper bound with intervention is squeezed by the intervention rate $\beta$. In practical algorithms, $\beta$ can be minimized to reduce the state distribution discrepancy and thus relieve the performance drop during test time. Based on Thm. 3.2, we further prove in Appendix A.1 that under the setting of intervention-based RL, the accumulated returns of behavior policy $J(\pi_b)$ and student policy $J(\pi_s)$ can be similarly related. The analysis in this section does not assume a certain form of the intervention function $\mathcal{T}(s)$. Our analysis provides the insight on the feasibility and efficiency of all previous algorithms in intervention-based RL (Kelly et al., 2019; Peng et al., 2021; Chisari et al., 2021). In the following section, we will examine different forms of intervention functions and investigate their properties and performance bounds, especially with imperfect online demonstrations.

## 3.2 LEARNING FROM IMPERFECT DEMONSTRATIONS

A straightforward idea to design the intervention function is to intervene when the student acts differently from the teacher. We model such process with the action-based intervention function $\mathcal{T}_{\text{action}}(s)$:

$$\mathcal{T}_{\text{action}}(s) = \begin{cases} 1 & \text{if } \mathbb{E}_{a \sim \pi_t(\cdot|s)}[\log \pi_s(a \mid s)] < \varepsilon, \\ 0 & \text{otherwise}, \end{cases} \tag{3}$$

wherein $\varepsilon > 0$ is a predefined parameter. A similar intervention function is used in EGPO (Peng et al., 2021), where the student's action is replaced by the teacher's if the student's action has low probability under the teacher's policy distribution. To measure the effectiveness of a certain form of intervention function, we examine the return of the behavior policy $J(\pi_b)$. With $\mathcal{T}_{\text{action}}(s)$ defined in Eq. 3 we can bound $J(\pi_b)$ with the following theorem.

**Theorem 3.3.** *With the action-based intervention function $\mathcal{T}_{\text{action}}(s)$, the return of the behavior policy $J(\pi_b)$ is lower and upper bounded by*

$$J(\pi_t) + \frac{\sqrt{2}(1 - \beta)R_{\max}}{(1 - \gamma)^2}\sqrt{H - \varepsilon} \geqslant J(\pi_b) \geqslant J(\pi_t) - \frac{\sqrt{2}(1 - \beta)R_{\max}}{(1 - \gamma)^2}\sqrt{H - \varepsilon}, \tag{4}$$

*where $H = \mathbb{E}_{s \sim d_{\pi_b}} \mathcal{H}(\pi_t(\cdot|s))$ is the average entropy of the teacher policy during shared control and $\beta$ is the weighted intervention rate in Thm. 3.2.*

The theorem shows that $J(\pi_b)$ can be lower bounded by the return of the teacher policy $\pi_t$ and an extra term relating to the entropy of the teacher policy. It implies that action-based intervention function $\mathcal{T}_{\text{action}}$ is indeed helpful in providing training data with high return. We discuss the tightness of Thm. 3.3 and give an intuitive interpretation of $\sqrt{H-\varepsilon}$ in Appendix A.2.

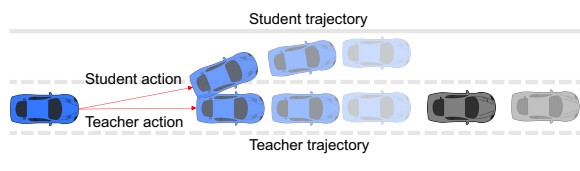

A drawback of the action-based intervention function is the strong assumption on the optimal teacher, which is not always feasible. If we turn to employ a suboptimal teacher, the behavior policy would be burdened due to the upper bound in Eq. 4. We illustrate this phenomenon with the example in Fig. 2 where a slow vehicle in gray is driving in front of the ego-vehicle in blue. The student policy is aggressive and would like to overtake the gray vehicle to reach the destination faster, while the teacher intends to follow the vehicle conservatively. Therefore, $\pi_s$ and $\pi_b$ will

Figure 2: In an autonomous driving scenario, the ego vehicle is the blue one on the left, following the gray vehicle on the right. The upper trajectory is proposed by the student to overtake and the lower trajectory is proposed by the teacher to keep following.

propose different actions in the current state, leading to $\mathcal{T}_{\text{action}} = 1$ according to Eq. 3. The mixed policy with shared control will always choose to follow the front vehicle and the agent can never accomplish a successful overtake.

To empower the student to outperform a suboptimal teacher policy, we investigate a new form of intervention function that encapsulates the long-term value estimation into the decision of intervention, designed as follows:

$$
\mathcal{T}_{\text{value}}(s) = \begin{cases} 1 & \text{if } V^{\pi_t}(s) - \mathbb{E}_{a\sim\pi_s(\cdot|s)}Q^{\pi_t}(s,a) > \varepsilon, \\ 0 & \text{otherwise,} \end{cases}
\tag{5}
$$

where $\varepsilon > 0$ is a predefined parameter. By using this intervention function, the teacher tolerates student's action if the teacher can not perform significantly better than the student by $\epsilon$ in return. $\mathcal{T}_{\text{value}}$ no longer expects the student to imitate the teacher policy step-by-step. Instead, it makes decision on the basis of long-term return. Taking trajectories in Fig. 2 again as an example, if the overtake behavior has high return, the student will be preferable to $\mathcal{T}_{\text{value}}$. Then the student control will not be intervened by the conservative teacher. So with the value-based intervention function, the agent's exploration ability will not be limited by a suboptimal teacher. Nevertheless, the lower-bound performance guarantee of the behavior policy $\pi_b$ still holds, shown as follows.

**Theorem 3.4.** *With the value-based intervention function $\mathcal{T}_{value}(s)$ defined in Eq. 5, the return of the behavior policy $\pi_b$ is lower bounded by*

$$
J(\pi_b) \geqslant J(\pi_t) - \frac{(1-\beta)\varepsilon}{1-\gamma}.
\tag{6}
$$

In safety-critical scenarios, the step-wise training cost $c(s,a)$, *i.e.*, the penalty on the safety violation during training, can be regarded as a negative reward. We define $\hat{r}(s,a) = r(s,a) - \eta c(s,a)$ as the combined reward, where $\eta$ is the weighting hyperparameter. $\hat{V}, \hat{Q}$ and $\hat{\mathcal{T}}_{\text{value}}$ are similarly defined by substituting $r$ with $\hat{r}$ in the original definition. Then we have the following corollary related to expected cumulative training cost, defined by $C(\pi) = \mathbb{E}_{s_0\sim d_0, a_t\sim\pi(\cdot|s_t), s_{t+1}\sim p(\cdot|s_t,a_t)}\left[\sum_{t=0}^{\infty}\gamma^t c(s_t,a_t)\right]$.

**Corollary 3.5.** *With safety-critical value-based intervention function $\hat{\mathcal{T}}_{value}(s)$, the expected cumulative training cost of the behavior policy $\pi_b$ is upper bounded by*

$$
C(\pi_b) \leqslant C(\pi_t) + \frac{(1-\beta)\epsilon}{\eta(1-\gamma)} + \frac{1}{\eta}\left[J(\pi_b) - J(\pi_t)\right].
\tag{7}
$$

In Eq. 7 the upper bound of behavior policy's training cost consists of three terms: the cost of teacher policy, the threshold in intervention $\epsilon$ multiplied by coefficients and the superiority of $\pi_b$ over $\pi_t$ in cumulative reward. The first two terms are similar to those in Eq. 6 and the third term means a trade-off between training safety and efficient exploration, which can be adjusted by hyperparameter $\eta$.

Comparing the lower bound performance guarantee of action-based and value-based intervention function (Eq. 4 and Eq. 6), the performance gap between $\pi_b$ and $\pi_t$ can both be bounded with respect to the threshold for intervention $\varepsilon$ and the discount factor $\gamma$. The difference is that the performance gap when using $\mathcal{T}_{\text{action}}$ is in an order of $O(\frac{1}{(1-\gamma)^2})$ while the gap with $\mathcal{T}_{\text{value}}$ is in an order of $O(\frac{1}{1-\gamma})$. It implies that in theory value-based intervention leads to better lower-bound performance guarantee. In terms of training safety guarantee, value-based intervention function $\mathcal{T}_{\text{value}}$ has better safety guarantee by providing a tighter safety bound with the order of $O(\frac{1}{1-\gamma})$, in contrast to $O(\frac{1}{(1-\gamma)^2})$ of action-based intervention function (see Theorem 1 in (Peng et al., 2021)). We show in the Sec. 4.3 that the theoretical advances of $\mathcal{T}_{\text{value}}$ in training safety and efficiency can both be verified empirically.

## 3.3 IMPLEMENTATION

Justified by the aforementioned advantages of the value-based intervention function, we propose a practical algorithm called **Teacher-Student Shared Control** (**TS2C**). Its workflow is listed in Appendix B. To obtain the teacher Q-network $Q^{\pi_t}$ in the value-based intervention function in Eq. 5, we rollout the teacher policy $\pi_t$ and collect training samples during the warmup period. Gaussian noise is added to the teacher's policy distribution to increase the state coverage during warmup. With limited training data the Q-network may fail to provide accurate estimation when encountering previously unseen states. We propose to use teacher Q-ensemble based on the idea of ensembling Q-networks (Chen et al., 2021a). A set of ensembled teacher Q-networks $\mathbf{Q}^\phi$ with the same architecture and different initialization weights are built and trained with the same data. To learn $\mathbf{Q}^\phi$ we follow the standard procedure in (Chen et al., 2021a) and optimize the following loss:

$$L(\phi) = \mathbb{E}_{s,a\sim\mathcal{D}} \left[ y - \text{Mean} \left[ \mathbf{Q}^\phi(s,a) \right] \right]^2, \tag{8}$$

where $y = \mathbb{E}_{s'\sim\mathcal{D},a'\sim\pi_t(\cdot|s')+\mathcal{N}(0,\sigma)} \left[ r + \gamma\text{Mean} \left[ \mathbf{Q}^\phi(s',a') \right] \right]$ is the Bellman target and $\mathcal{D}$ is the replay buffer for storing sequences $\{(s,a,r,s')\}$. Teacher will intervene when $\mathcal{T}_{\text{value}}$ returns 1 or the output variance of ensembled Q-networks surpasses the threshold, which means the agent is exploring unknown regions and requires guarding. We also use $\mathbf{Q}^\phi$ to compute the state-value functions in Eq. 5, leading to the following practical intervention function:

$$\mathcal{T}_{\text{TS2C}}(s) = \begin{cases} 1 & \text{if } \text{Mean} \left[ \mathbb{E}_{a\sim\pi_t(\cdot|s)}\mathbf{Q}^\phi(s,a) - \mathbb{E}_{a\sim\pi_s(\cdot|s)}\mathbf{Q}^\phi(s,a) \right] > \varepsilon_1 \\ & \text{or } \text{Var} \left[ \mathbb{E}_{a\sim\pi_s(\cdot|s)}\mathbf{Q}^\phi(s,a) \right] > \varepsilon_2, \\ 0 & \text{otherwise.} \end{cases} \tag{9}$$

Eq. 2 shows that the distributional shift and the performance gap to oracle can be reduced with smaller $\beta$, i.e., less teacher intervention. Therefore, we minimize the amount of teacher intervention via adding negative reward to the transitions one step before the teacher intervention. Incorporating intervention minimization, we use the following loss function to update the student's Q-network parameterized by $\psi$:

$$L(\psi) = \mathbb{E}_{s,a\sim\mathcal{D}} \left[ \left( y' - Q^\psi(s,a) \right)^2 \right], \tag{10}$$

where $y' = \mathbb{E}_{s'\sim\mathcal{D},a'\sim\pi_b(\cdot|s')} \left[ r - \lambda\mathcal{T}_{\text{TS2C}}(s') + \gamma Q^\psi(s',a') - \alpha\log\pi_b(a'|s') \right]$ is the soft Bellman target with intervention minimization. $\lambda$ is the hyperparameter controlling the intervention minimization. $\alpha$ is the coefficient for maximum-entropy learning updated in the same way as Soft Actor Critic (SAC) (Haarnoja et al., 2018). To update the student's policy network parameterized by $\theta$, we apply the objective used in SAC as:

$$L(\theta) = \mathbb{E}_{s\sim\mathcal{D}} \left[ \mathbb{E}_{a\sim\pi^\theta(\cdot|s)} \left[ \alpha\log\left(\pi^\theta(a\mid s)\right) - Q^\psi(s,a) \right] \right]. \tag{11}$$

## 4 EXPERIMENTS

We conduct experiments to investigate the following questions: (1) Can agents trained with TS2C achieve super-teacher performance with imperfect teacher policies while outperforming other methods in the Teacher-Student Framework (TSF)? (2) Can TS2C provide safety guarantee and improve training efficiency compared to algorithms without teacher intervention? (3) Is TS2C robust in different environments and teacher policies trained with different algorithms? To answer questions

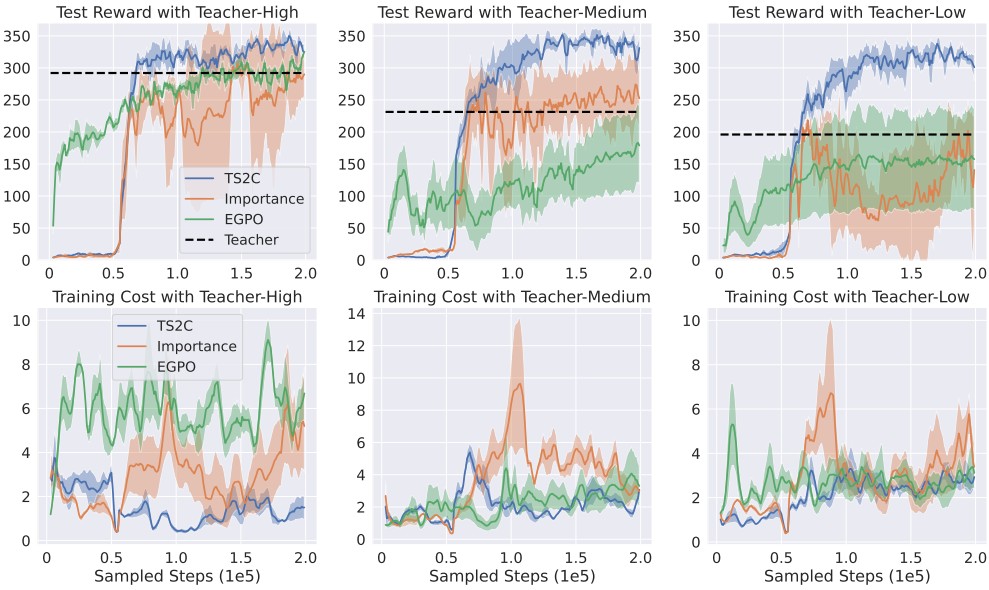

Figure 3: Comparison between our method TS2C and other algorithms with teacher policies providing online demonstrations. "Importance" refers to the Importance Advising algorithm. For each column, the involved teacher policy has high, medium, and low performance respectively.

(1)(2), we conduct preliminary training with the PPO algorithm (Schulman et al., 2017) and save checkpoints on different timesteps. Policies in different stages of PPO training are used as teacher policies in TS2C and other algorithms in the TSF. With regard to question (3), we use agents trained with PPO (Schulman et al., 2017), SAC (Haarnoja et al., 2018) and Behavior Cloning as the teacher policies from different sources.

## 4.1 ENVIRONMENT SETUP

The majority of the experiments are conducted on the lightweight driving simulator MetaDrive (Li et al., 2022a). One concern with TSF algorithms is that the student may simply record the teacher's actions and overfit the training environment. MetaDrive can test the generalizability of learned agents on unseen driving environments with its capability to generate an unlimited number of scenes with various road networks and traffic flows. We choose 100 scenes for training and 50 held-out scenes for testing. Examples of the traffic scenes from MetaDrive are shown in Appendix C. In MetaDrive, the objective is to drive the ego vehicle to the destination without dangerous behaviors such as crashing into other vehicles. The reward function consists of the dense reward proportional to the vehicle speed and the driving distance, and the terminal +20 reward when the ego vehicle reaches the destination. Training cost is increased by 1 when the ego vehicle crashes or drives out of the lane. To evaluate TS2C's performance in different environments, we also conduct experiments in several environments of the MuJoCo simulator (Todorov et al., 2012).

## 4.2 BASELINES AND IMPLEMENTATION DETAILS

Two sets of algorithms are selected as baselines to compare with. One includes traditional RL and IL algorithms without the TSF. By comparing with these methods we can demonstrate how TS2C improves the efficiency and safety of training. Another set contains previous algorithms with the TSF, including Importance Advising (Torrey & Taylor, 2013) and EGPO (Peng et al., 2021). The original Importance Advising uses an intervention function based on the range of the Q-function: $I(s) = \max_{a \in A} Q_{\mathcal{D}(s,a)} - \min_{a \in A} Q_{\mathcal{D}(s,a)}$, where $Q_{\mathcal{D}}$ is the Q-table of the teacher policy. Such Q-table is not applicable in the Metadrive simulator with continuous state and action spaces. In practice, we sample $N$ actions from the teacher's policy distribution and compute their Q-values on a certain state. The intervention happens if the range, i.e., the maximum value minus the minimum

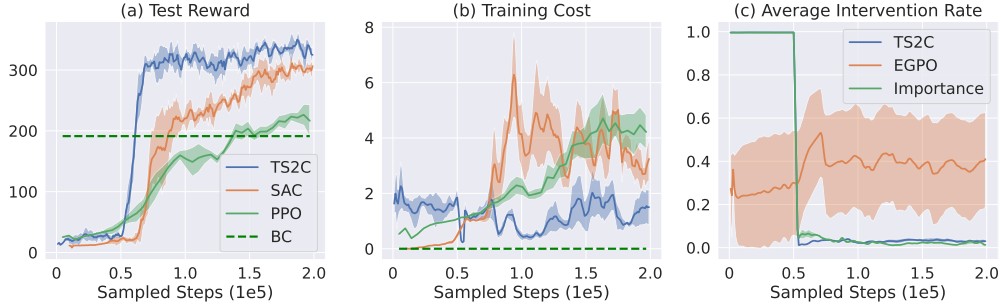

Figure 4: Figures (a) and (b) shows the comparison of efficiency and safety between TS2C and baseline algorithms without teacher policies providing online demonstrations. Figure (c) shows the comparison of the average intervention rate between TS2C and two baseline algorithms in the TSF.

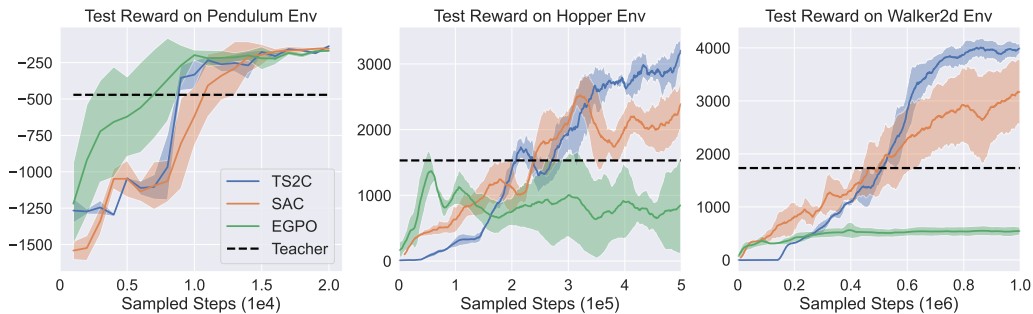

Figure 5: Performance comparison between our method TS2C and baseline algorithms on three environments from MuJoCo.

value, surpass a certain threshold $\varepsilon$. The EGPO algorithm uses an intervention function similar to the action-based intervention function introduced in section 3.2. All algorithms are trained with 4 different random seeds. In all figures the solid line is computed with the average value across different seeds and the shadow implies the standard deviation. We leave detailed information on the experiments and the result of ablation studies on hyperparameters in Appendix C.

## 4.3 RESULTS

**Super-teacher performance and better safety guarantee** The training result with three different levels of teacher policy can be seen in Fig. 3. The first row shows that the performance of TS2C is not limited by the imperfect teacher policies. It converges within 200k steps, independent of different performances of the teacher. EGPO and Importance Advicing is clearly bounded by teacher-medium and teacher-low, performing much worse than TS2C with imperfect teachers. The second row of Fig. 3 shows TS2C has lower training cost than both algorithms. Compared to EGPO and Importance Advising, the intervention mechanism in TS2C is better-designed and leads to better behaviors.

**Better performance with TSF** The result of comparing TS2C with baseline methods without the TSF can be seen in Fig. 4(a)(b). We use the teacher policy with a medium level of performance to train the student in TS2C. It achieves better performance and lower training cost than the baseline algorithms SAC, PPO, and BC. The comparative results show the effectiveness of incorporating teacher policies in online training. The behavior cloning algorithm does not involve online sampling in the training process, so it has zero training cost.

**Extension for different environments and teacher policies** The performances of TS2C in different MuJoCo environments and different sources of teacher policy are presented in Fig. 5 and 6 respectively. The figures show that TS2C is generalizable to different environments. It can also make use of the teacher policies from different sources and achieve super-teacher performance consistently. Our TS2C algorithm can outperform SAC in all three MuJoCo environments taken into consideration. On the other hand, though the EGPO algorithm has the best performance in the Pendulum environment, it struggles in the other two environments, namely Hopper and Walker.

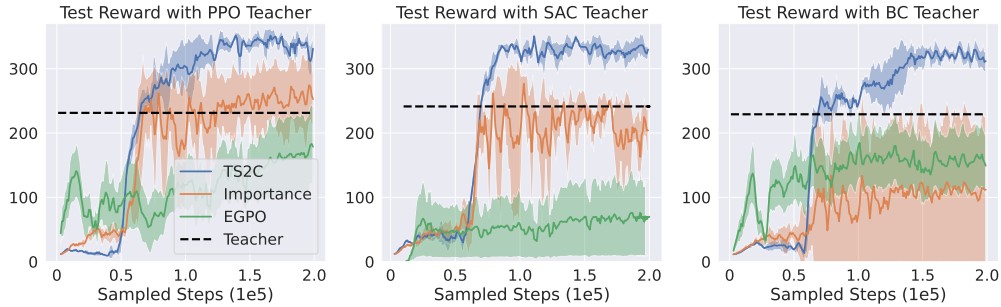

Figure 6: Performance comparison between our method TS2C and baseline algorithms with teacher policies providing online demonstrations. The teacher policies are trained by PPO, SAC, and behavior cloning respectively.

## 4.4 Effects of Intervention Functions

We further investigate the intervention behaviors under different intervention functions. As shown in Fig. 4(c), the average intervention rate $\mathbb{E}_{s \sim d_{\pi_b}} \mathcal{T}(s)$ of TS2C drops quickly as soon as the student policy takes control. The teacher policy only intervenes during a very few states where it can propose actions with higher value than the students. The intervention rate of EGPO remains high due to the action-based intervention function: the teacher intervenes whenever the student act differently.

We also show different outcomes of action-based and value-based intervention functions with screenshots in the MetaDrive simulator. In Fig. 7 the ego vehicle happens to drive behind a traffic vehicle which is in an orange trajectory. With action-based intervention the teacher takes control and keeps following the front vehicle, as shown in the green trajectory. In contrast, with the value-based intervention the student policy proposes to turn left and overtake the front vehicle as in the blue trajectory. Such action has higher return and therefore is tolerable by $\mathcal{T}_{\text{TS2C}}$, leading to a better agent trajectory.

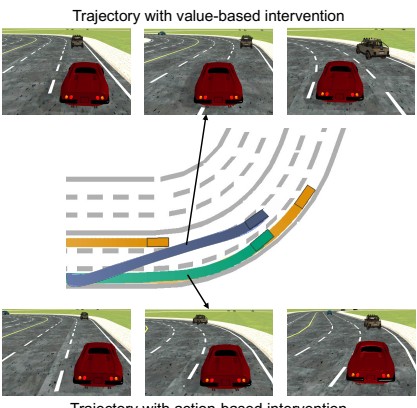

Figure 7: Visualization of the trajectories resulting from different intervention mechanisms. The trajectories of irrelevant traffic vehicles are marked orange. As in the green trajectory, action-based intervention make the car following the front vehicle. Value-based intervention instead can learn overtaking behavior as in blue trajectory.

## 5 Conclusion and Discussion

In this work, we conduct theoretic analysis on intervention-based RL algorithms in the Teacher-Student Framework. It is found that while the intervention mechanism has better properties than some imitation learning methods, using an action-based intervention function limits the performance of the student policy. We then propose TS2C, a value-based intervention scheme for online policy optimization with imperfect teachers. We provide the theoretic guarantees on its exploration ability and safety. Experiments show that the proposed TS2C method achieves consistent performance independent to the teacher policy being used. Our work brings progress and potential impact to relevant topics such as active learning, human-in-the-loop methods, and safety-critical applications.

**Limitations.** The proposed algorithm assumes the agent can access environment rewards, and thus defines the intervention function based on value estimations. It may not work in tasks where reward signals are inaccessible. This limitation could be tackled by considering reward-free settings and employing unsupervised skill discovery (Eysenbach et al., 2019; Aubret et al., 2019). These methods provide proxy reward functions that can be used in teacher intervention.

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

# A    THEOREMS IN TS2C

## A.1    DETAILED PROOF

We start the proof with the restatement of Lem. 3.1 in Sec. 3.1.

**Lemma A.1** (Lemma 4.1 in (Xu et al., 2019))**.**

$$\|d_\pi - d_{\pi'}\|_1 \leqslant \frac{\gamma}{1-\gamma} \mathbb{E}_{s \sim d_\pi} \|\pi(\cdot \mid s) - \pi'(\cdot \mid s)\|_1. \tag{12}$$

Thm 3.2 can be derived by substituting $\pi$ and $\pi'$ in Lem A.1 with $\pi_b$ and $\pi_s$.

**Theorem A.2** (Restatement of Thm. 3.2)**.** *For any behavior policy $\pi_b$ deduced by a teacher policy $\pi_t$, a student policy $\pi_s$ and a intervention function $\mathcal{T}(s)$, the state distribution discrepancy between $\pi_b$ and $\pi_s$ is bounded by policy discrepancy and intervention rate:*

$$\|d_{\pi_b} - d_{\pi_s}\|_1 \leqslant \frac{\beta\gamma}{1-\gamma} \mathbb{E}_{s \sim d_{\pi_b}} \|\pi_t(\cdot \mid s) - \pi_s(\cdot \mid s)\|_1, \tag{13}$$

*where $\beta = \frac{\mathbb{E}_{s \sim d_{\pi_b}} \|\mathcal{T}(s)[\pi_t(\cdot|s) - \pi_s(\cdot|s)]\|_1}{\mathbb{E}_{s \sim d_{\pi_b}} \|\pi_t(\cdot|s) - \pi_s(\cdot|s)\|_1}$ is the weighted expected intervention rate.*

*Proof.*

$$\begin{aligned}
\|d_{\pi_b} - d_{\pi_s}\|_1 &\leqslant \frac{\gamma}{1-\gamma} \mathbb{E}_{s \sim d_{\pi_b}} \|\pi_b(\cdot \mid s) - \pi_s(\cdot \mid s)\|_1 \\
&= \frac{\gamma}{1-\gamma} \mathbb{E}_{s \sim d_{\pi_b}} \|\mathcal{T}(s)\pi_t(\cdot \mid s) + (1 - \mathcal{T}(s))\pi_s(\cdot \mid s) - \pi_s(\cdot \mid s)\|_1 \\
&= \frac{\gamma}{1-\gamma} \mathbb{E}_{s \sim d_{\pi_b}} \|\mathcal{T}(s)[\pi_t(\cdot \mid s) - \pi_s(\cdot \mid s)]\|_1 \\
&= \frac{\beta\gamma}{1-\gamma} \mathbb{E}_{s \sim d_{\pi_b}} \|\pi_t(\cdot \mid s) - \pi_s(\cdot \mid s)\|_1.
\end{aligned} \tag{14}$$

$\square$

Based on Thm. 3.2, we further prove that under the setting of shared control, the performance gap of $\pi_s$ to the optimal policy $\pi^*$ can be bounded by the gap between the teacher policy $\pi_t$ and $\pi^*$, together with the teacher-student policy difference. Therefore, training with the trajectory collected with mixed policy $\pi_b$ is to optimize an upper bound of the student's suboptimality. The following lemma is helpful in doing this.

**Lemma A.3.**

$$|J(\pi) - J(\pi')| \leqslant \frac{R_{\max}}{(1-\gamma)^2} \mathbb{E}_{s \sim d_\pi} \|\pi(\cdot \mid s) - \pi'(\cdot \mid s)\|_1 \tag{15}$$

*Proof.* It is a direct combination of Lemma 4.2 and Lemma 4.3 in (Xu et al., 2019). $\square$

**Theorem A.4.** *For any behavior policy $\pi_b$ consisting of a teacher policy $\pi_t$, a student policy $\pi_s$ and a intervention function $\mathcal{T}(s)$, the suboptimality of the student policy is bounded by*

$$|J(\pi^*) - J(\pi_s)| \leqslant \frac{\beta R_{\max}}{(1-\gamma)^2} \mathbb{E}_{s \sim \pi_b} \|\pi_t(\cdot \mid s) - \pi_s(\cdot \mid s)\|_1 + |J(\pi^*) - J(\pi_b)|, \tag{16}$$

*Proof.*

$$
\begin{aligned}
|J\left(\pi_b\right)-J\left(\pi_s\right)| &\leqslant \frac{R_{\max}}{(1-\gamma)^2}\mathbb{E}_{s\sim d_{\pi_b}}\left\|\pi_b(\cdot\mid s)-\pi_s(\cdot\mid s)\right\|_1 \\
&= \frac{R_{\max}}{(1-\gamma)^2}\mathbb{E}_{s\sim d_{\pi_b}}\left\|\mathcal{T}(s)\pi_t(\cdot\mid s)+(1-\mathcal{T}(s))\pi_s(\cdot\mid s)-\pi_s(\cdot\mid s)\right\|_1 \\
&= \frac{R_{\max}}{(1-\gamma)^2}\mathbb{E}_{s\sim d_{\pi_b}}\left\|\mathcal{T}(s)\left[\pi_t(\cdot\mid s)-\pi_s(\cdot\mid s)\right]\right\|_1 \\
&= \frac{\beta R_{\max}}{(1-\gamma)^2}\mathbb{E}_{s\sim\pi_b}\left\|\pi_t(\cdot\mid s)-\pi_s(\cdot\mid s)\right\|_1.
\end{aligned}
\tag{17}
$$
$$
\begin{aligned}
|J\left(\pi^*\right)-J\left(\pi_s\right)| &\leqslant |J\left(\pi_b\right)-J\left(\pi_s\right)|+|J\left(\pi^*\right)-J\left(\pi_b\right)| \\
&\leqslant \frac{\beta R_{\max}}{(1-\gamma)^2}\mathbb{E}_{s\sim\pi_b}\left\|\pi_t(\cdot\mid s)-\pi_s(\cdot\mid s)\right\|_1+|J\left(\pi^*\right)-J\left(\pi_b\right)|.
\end{aligned}
$$

$\square$

**Theorem A.5** (Restatement of Thm. 3.3). *With the action distributional intervention function $\mathcal{T}_{action}(s)$, the return of the behavior policy $J(\pi_b)$ is lower and upper bounded by*

$$
J(\pi_t)+\frac{\sqrt{2}(1-\beta)R_{\max}}{(1-\gamma)^2}\sqrt{H-\varepsilon} \geqslant J(\pi_b) \geqslant J(\pi_t)-\frac{\sqrt{2}(1-\beta)R_{\max}}{(1-\gamma)^2}\sqrt{H-\varepsilon}
\tag{18}
$$

*where $R_{\max}=\max\limits_{s,a} r(s,a)$ is the maximal possible reward, $H=\mathbb{E}_{s\sim d_{\pi_b}}\mathcal{H}(\pi^t(\cdot|s))$ is the average entropy of the teacher policy during shared control.*

*Proof.*

$$
\begin{aligned}
|J\left(\pi_b\right)-J\left(\pi_t\right)| &\leqslant \frac{R_{\max}}{(1-\gamma)^2}\mathbb{E}_{s\sim d_{\pi_b}}\left\|\pi_b(\cdot\mid s)-\pi_t(\cdot\mid s)\right\|_1 \\
&= \frac{R_{\max}}{(1-\gamma)^2}\mathbb{E}_{s\sim d_{\pi_b}}\left\|\mathcal{T}(s)\pi_t(\cdot\mid s)+(1-\mathcal{T}(s))\pi_s(\cdot\mid s)-\pi_t(\cdot\mid s)\right\|_1 \\
&= \frac{(1-\beta)R_{\max}}{(1-\gamma)^2}\mathbb{E}_{s\sim d_{\pi_b}}\left\|\pi_s(\cdot\mid s)-\pi_t(\cdot\mid s)\right\|_1 \\
&\leqslant \frac{\sqrt{2}(1-\beta)R_{\max}}{(1-\gamma)^2}\mathbb{E}_{s\sim d_{\pi_b}}\sqrt{\mathrm{D}_{\mathrm{KL}}(\pi_t(\cdot|s)\|\pi_s(\cdot|s))} \\
&= \frac{\sqrt{2}(1-\beta)R_{\max}}{(1-\gamma)^2}\mathbb{E}_{s\sim d_{\pi_b}}\sqrt{\mathbb{E}_{a\sim\pi_t(\cdot|s)}\left[\log\pi_t(a|s)-\log\pi_s(a|s)\right]} \\
&= \frac{\sqrt{2}(1-\beta)R_{\max}}{(1-\gamma)^2}\mathbb{E}_{s\sim d_{\pi_b}}\sqrt{\mathcal{H}(\pi^t(\cdot|s)-\varepsilon} \\
&\leqslant \frac{\sqrt{2}(1-\beta)R_{\max}}{(1-\gamma)^2}\sqrt{H-\varepsilon}.
\end{aligned}
\tag{19}
$$

Therefore, we obtain

$$
\begin{aligned}
\frac{\sqrt{2}(1-\beta)R_{\max}}{(1-\gamma)^2}\sqrt{H-\varepsilon} &\geqslant J\left(\pi_b\right)-J\left(\pi_t\right) \geqslant -\frac{\sqrt{2}(1-\beta)R_{\max}}{(1-\gamma)^2}\sqrt{H-\varepsilon} \\
J(\pi_t)+\frac{\sqrt{2}(1-\beta)R_{\max}}{(1-\gamma)^2}\sqrt{H-\varepsilon} &\geqslant J(\pi_b) \geqslant J(\pi_t)-\frac{\sqrt{2}(1-\beta)R_{\max}}{(1-\gamma)^2}\sqrt{H-\varepsilon},
\end{aligned}
\tag{20}
$$

which concludes the proof. $\square$

To prove Thm. 3.4, we introduce a useful lemma from (Schulman et al., 2015).

**Lemma A.6.**

$$
J(\pi)=J(\pi')+\mathbb{E}_{s_t,a_t\sim\tau_\pi}\left[\sum_{t=0}^{\infty}\gamma^t A_{\pi'}\left(s_t,a_t\right)\right]
\tag{21}
$$

**Theorem A.7** (Restatement of Thm. 3.4). *With the value-based intervention function $\mathcal{T}_{value}(s)$ defined in Eq. 5, the return of the behavior policy $\pi_b$ is lower bounded by*

$$J(\pi_b) \geqslant J(\pi_t) - \frac{\varepsilon}{1-\gamma}. \tag{22}$$

*Proof.*

$$
\begin{aligned}
J(\pi_b) - J(\pi_t) &= \mathbb{E}_{s_n,a_n \sim \tau_{\pi_b}} \left[ \sum_{n=0}^{\infty} \gamma^n A_t(s_n, a_n) \right] \\
&= \mathbb{E}_{s_n,a_n \sim \tau_{\pi_b}} \left[ \sum_{n=0}^{\infty} \gamma^n \left[ Q_t(s_n, a_n) - V_t(s_n) \right] \right] \\
&= \mathbb{E}_{s_n \sim \tau_{\pi_b}} \left[ \sum_{n=0}^{\infty} \gamma^n \left[ \mathbb{E}_{a \sim \pi_b(\cdot|s_n)} Q_t(s_n, a) - V_t(s_n) \right] \right] \\
&= \mathbb{E}_{s_n \sim \tau_{\pi_b}} \left[ \sum_{n=0}^{\infty} \gamma^n \left[ \mathcal{T}(s_n) \mathbb{E}_{a \sim \pi_t(\cdot|s_n)} Q_t(s_n, a) + (1 - \mathcal{T}(s_n)) \mathbb{E}_{a \sim \pi_s(\cdot|s_n)} Q_t(s_n, a) - V_t(s_n) \right] \right] \\
&= \mathbb{E}_{s_n \sim \tau_{\pi_b}} \left[ \sum_{n=0}^{\infty} \gamma^n \left[ (1 - \mathcal{T}(s_n)) \left[ \mathbb{E}_{a \sim \pi_s(\cdot|s_n)} Q_t(s_n, a) - V_t(s_n) \right] \right] \right] \\
&= (1 - \beta) \mathbb{E}_{s_n \sim \tau_{\pi_b}} \left[ \sum_{n=0}^{\infty} \gamma^n \left[ \mathbb{E}_{a \sim \pi_s(\cdot|s_n)} Q_t(s_n, a) - V_t(s_n) \right] \right] \\
&\geqslant -(1 - \beta) \mathbb{E}_{s_n \sim \tau_{\pi_b}} \left[ \sum_{n=0}^{\infty} \gamma^n \varepsilon \right] \\
&= -\frac{(1 - \beta)\varepsilon}{1 - \gamma},
\end{aligned}
\tag{23}
$$

which concludes the proof. $\qquad\square$

Then we prove the corollary related to safety-critical scenarios.

**Corollary A.8** (Restatement of Cor. 3.5). *With safety-critical value-based intervention function $\hat{\mathcal{T}}_{value}(s)$, the expected cumulative training cost of the behavior policy $\pi_b$ is upper bounded by*

$$C(\pi_b) \leqslant C(\pi_t) + \frac{(1-\beta)\epsilon}{\eta(1-\gamma)} + \frac{1}{\eta} \left[ J(\pi_b) - J(\pi_t) \right]. \tag{24}$$

*Proof.* We define expected return under policy $\pi$ with combined reward $\hat{r}$ as $\hat{J}(\pi)$, therefore

$$
\begin{aligned}
\hat{J}(\pi) &= \mathbb{E}_{s_0 \sim d_0, a_t \sim \pi(\cdot|s_t), s_{t+1} \sim p(\cdot|s_t, a_t)} \left[ \sum_{t=0}^{\infty} \gamma^t \hat{r}(s_t, a_t) \right] \\
&= \mathbb{E}_{s_0 \sim d_0, a_t \sim \pi(\cdot|s_t), s_{t+1} \sim p(\cdot|s_t, a_t)} \left[ \sum_{t=0}^{\infty} \gamma^t \left[ r(s_t, a_t) + \eta c(s_t, a_t) \right] \right] \\
&= J(\pi) + \eta C(\pi)
\end{aligned}
\tag{25}
$$

According to Thm. A.7, under $\hat{T}_{\text{value}}(s)$ we have

$$\hat{J}(\pi_b) \geqslant \hat{J}(\pi_t) - \frac{\varepsilon}{1-\gamma}. \tag{26}$$

Eq. 24 can be immediately proved by combining Eq. 25 and Eq. 26. $\qquad\square$

A.2    DISCUSSIONS ON THE RESULTS

In Thm. 3.3, the average entropy of the teacher policy $H$ and the threshold for action-based intervention $\varepsilon$ is included in the bound. We provide intuitive interpretations on the influence of $H$ and $\varepsilon$ here. For reference, the action-based intervention function $T_{action} = 1$ when $\mathbb{E}_{a \sim \pi_t(\cdot|s)} [\log \pi_s(a \mid s)] < \varepsilon$. According to Thm 3.3 of our paper, a larger $\varepsilon$ leads to smaller discrepancy between the returns of the behavior and teacher policies. This is because $\varepsilon$ is the threshold for the action-based intervention function. If the action likelihood is less than $\varepsilon$, the teacher policy will take over the control. A larger $\varepsilon$ means more teacher intervention, constraining the behavior policy to be closer to the teacher policy, which leads to a smaller discrepancy in their returns. The influence of $H$ can be similarly analyzed. A larger $H$ leads to larger return discrepancy. Intuitively, this is because with higher entropy, the teacher policy tends to have a more "averaged" or multi-modal distribution over the action space. So the policy distributions of the student and teacher are more likely to have overlaps, leading to a higher action likelihood. In turn, the intervention criterion is less likely to be satisfied, leading to fewer teacher interventions. In general, the intuitive interpretation of Thm. 3.3 indicates that if we would like larger return discrepancy, i.e. larger performance upper bound as well as smaller lower bound, we should use smaller intervention threshold and teacher policy with higher entropy, and vice versa. Thm. 3.3 has a gap with the actual algorithm in that the algorithm uses a value-based intervention function which is based on Thm. 3.4. Nevertheless, the intuitive interpretation may enlighten future work on how to choose a proper teacher policy in teacher-student shared control.

With respect to the tightness, Thm. 3.3 has a squared planning horizon $\frac{1}{(1-\gamma)^2}$ in the discrepancy term. This is in accordance with many previous works (Thm. 1 in (Xu et al., 2019), Thm. 4.1 in (Janner et al., 2019) and Thm. 1 in (Schulman et al., 2015)), which include $(1 - \gamma)^2$ in the denominator when it comes to differences of the cumulative return, given the difference in the action distribution. The order of $\frac{1}{1-\gamma}$ in Thm. 3.3 is tight, which dominates the gap in accumulated return. Nevertheless, the other constant terms, e.g. $R_{\max}$ and the average entropy, can be tighter given some additional assumptions. We did not derive a tighter bound since the derivation will not be related to the main contribution of this paper, which is the new type of intervention function. Thm. 3.3 and Thm. 3.4 in their current forms are enough to demonstrate that the value-based intervention function has the advantage of providing more efficient exploration and better safety guarantee compared with action-based intervention function.

# B    THE ALGORITHM

The workflow of TS2C during training is show in Alg. 1.

---

**Algorithm 1** The workflow of TS2C during training

---

1: **Input:** Warmup steps $W$; Scale of warmup noise $\sigma$; Training steps $N$; Teacher policy $\pi_t$.
2: Initialize student policy $\pi_s^\theta$, a set of parameterized Q-function for teacher policy $\mathbf{Q}^\phi$, parameterized Q-function for student policy $Q^\psi$ and the replay buffer $D$.
3: **for** $i = 1$ **to** $W$ **do**
4:     Observe state $s_i$ and sample $a_i \sim \pi_t(\cdot|s) + \mathcal{N}(0, \sigma)$.
5:     Step the environment with $a_i$ and store the tuple $(s_i, a_i, r_i, s_{i+1})$ to $D$.
6:     Update $\phi$ with Temporal-Difference loss in Eq. 8.
7: **end for**
8: **for** $i = 1$ **to** $N$ **do**
9:     Observe state $s_i$ and sample $a_t \sim \pi_t(\cdot|s_i)$, $a_s \sim \pi_s^\theta(\cdot|s_i)$.
10:     Compute $\mathcal{T}_{\text{ts2c}}(s_i)$ with Eq. 9, behavior policy $\pi_b(\cdot|s_i)$ and $a_b$.
11:     Step the environment with $a_b$ and store the tuple $(s_i, a_b, r_i, s_{i+1}, \mathcal{T}_{\text{value}}(s_{i+1}))$ to $D$.
12:     Update $\psi$ in the student Q-function with the loss in Eq. 10.
13:     Update $\theta$ in the student policy with the loss in Eq. 11.
14: **end for**

---

## C ADDITIONAL EXPERIMENT DEMONSTRATIONS

### C.1 DEMONSTRATIONS OF DRIVING SCENARIOS

The demonstrations of several driving scenarios are shown in Fig. 8. We provide a demonstrative video showing the agent behavior trained with PPO and our TS2C algorithm in the supplementary materials.

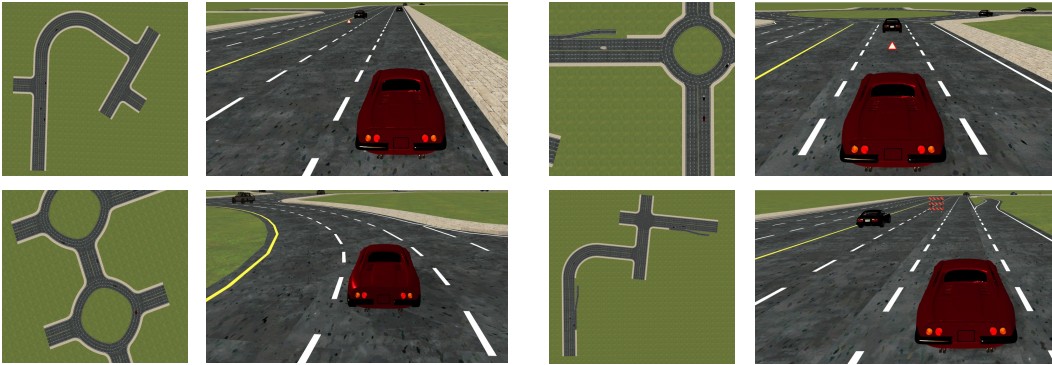

Figure 8: Four examples of the traffic scenes in MetaDrive.

### C.2 HYPER-PARAMETERS

The hyper-parameters used in the experiments are shown in the following tables. In the TS2C algorithm, larger values of the intervention threshold $\varepsilon_1$ and $\varepsilon_2$ will lead to a more strict intervention criterion and the steps with teacher control will be fewer. In order to control the policy distribution discrepancy, we choose $\varepsilon_1$ and $\varepsilon_2$ to ensure the average intervention rate to be less than 5%. Nevertheless, different $\varepsilon_1$ in the intervention function has little influence on the algorithm performance, as shown in Fig. 11 of our paper. The coefficient for intervention minimization $\lambda$ is simply set to 1. If used in other environments, it may need some adjustments to fit the reward scale. The coefficient for maximum entropy learning $\alpha$ is updated during training as in the SAC algorithm. The number of warmup timesteps is empirically chosen so that the expert value function can be properly trained. Other parameters follow the setting in EGPO (Peng et al., 2021). The hyper-parameters of other algorithms follow their original setting.

Table 1: TS2C (Ours)

| Hyper-parameter | Value |
|---|---|
| Discount Factor $\gamma$ | 0.99 |
| $\tau$ for target network update | 0.005 |
| Learning Rate | 0.0001 |
| Environmental horizon $T$ | 2000 |
| Warmup Timesteps $W$ | 50000 |
| # of Ensembled Value-Functions $N$ | 10 |
| Variance of Gaussian Noise $C$ | 0.5 |
| Intervention Minimization Ratio $\lambda$ | 1 |
| Value-based Intervention Threshold $\varepsilon_1$ | 1.2 |
| Value-based Intervention Threshold $\varepsilon_2$ | 2.5 |
| Activation Function | Relu |
| Hidden Layer Sizes | [256, 256] |

Table 2: EGPO (Peng et al., 2021)

| Hyper-parameter | Value |
|---|---|
| Discount Factor $\gamma$ | 0.99 |
| $\tau$ for target network update | 0.005 |
| Learning Rate | 0.0001 |
| Environmental horizon $T$ | 2000 |
| Steps before Learning start | 10000 |
| Intervention Occurrence Limit $C$ | 20 |
| Number of Online Evaluation Episode | 5 |
| $K_p$ | 5 |
| $K_i$ | 0.01 |
| $K_d$ | 0.1 |
| CQL Loss Temperature $\beta$ | 3.0 |
| Activation Function | Relu |
| Hidden Layer Sizes | [256, 256] |

Table 3: Importance Advising (Torrey & Taylor, 2013)

| Hyper-parameter | Value |
|---|---|
| Discount Factor $\gamma$ | 0.99 |
| $\tau$ for target network update | 0.005 |
| Learning Rate | 0.0001 |
| Environmental horizon $T$ | 2000 |
| Warmup Timesteps $W$ | 50000 |
| # of Actions Sampled $N$ | 10 |
| Variance of Gaussian Noise $C$ | 0.5 |
| Range-based Intervention Threshold $\varepsilon$ | 2.8 |
| Activation Function | Relu |
| Hidden Layer Sizes | [256, 256] |

Table 4: SAC (Haarnoja et al., 2018)

| Hyper-parameter | Value |
|---|---|
| Discount Factor $\gamma$ | 0.99 |
| $\tau$ for Target Network Update | 0.005 |
| Learning Rate | 0.0001 |
| Environmental Horizon $T$ | 2000 |
| Steps before Learning starts | 10000 |
| Activation Function | Relu |
| Hidden Layer Sizes | [256, 256] |

# D   ADDITIONAL EXPERIMENT RESULTS

## D.1   ADDITIONAL PERFORMANCE COMPARISONS ON METADRIVE

In Fig. 9, we show the results of TS2C trained with various levels of teachers compared with baseline algorithms without shared control. Apart from the Fig. 4 in the main paper presenting the training results of TS2C with the medium level of teacher policy, here we present the performance of TS2C trained with the high, medium and low levels of teacher policy. The value-based intervention proposed by TS2C can utilize all these teacher policies, leading to safer and more efficient training compared to traditional RL algorithms.

Fig. 10 shows the results with different levels of teacher policy. Besides the testing reward and the training cost shown in Fig. 3 of the main paper, we show the training reward and test success rate of TS2C compared with baseline methods with the Teacher-Student Framework (TSF) respectively. Our TS2C algorithm still achieves the best performance among baseline algorithms when evaluated with these two metrics.

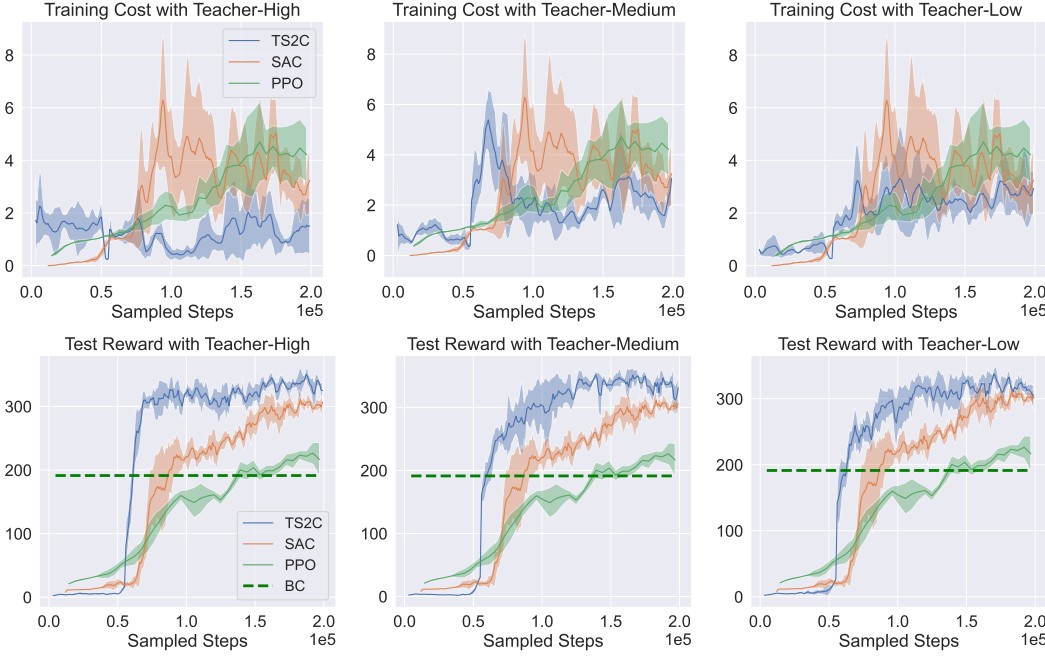

Figure 9: Comparison of training cost and test reward between our method TS2C and other algorithms without shared control.

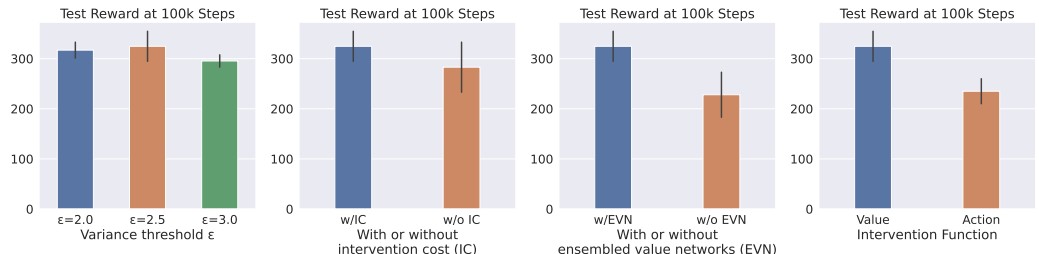

Figure 11: Ablation Studies for different variance thresholds, the intervention cost, ensembled value networks and different intervention functions.

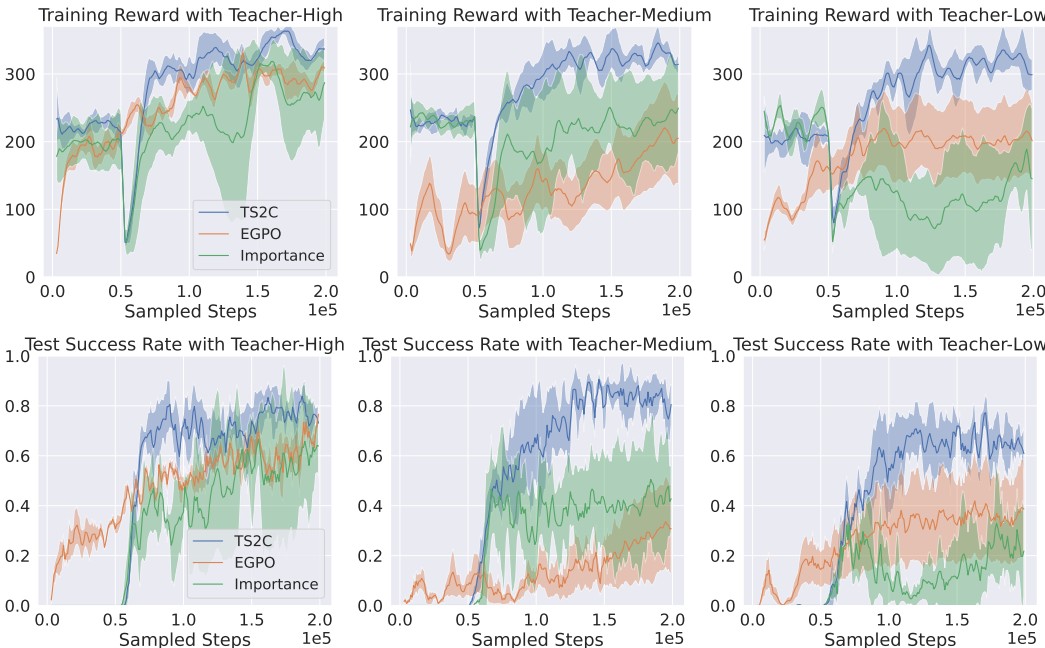

Figure 10: Comparison of training reward and test success rate between our method TS2C and other algorithms with shared control.

## D.2 ABLATION STUDIES

We conduct ablation studies and present the results in Fig. 11. We find the intervention cost and ensembled value networks are important to the algorithm's performance, while different variance thresholds in the intervention function has little influence. Also, TS2C with action-based intervention function behaves poorly in accordance with the theoretical analysis in Section 3.2.

## D.3 DISCUSSIONS ON EXPERIMENT RESULTS

In Fig. 5, our TS2C algorithm can outperform SAC in all three MuJoCo environments taken into consideration. On the other hand, though the EGPO algorithm has the best performance in the Pendulum environment, it struggles in the other two environments, namely Hopper and Walker. This is because the action space of the pendulum environment is only one-dimensional. In this simple environment, the action-based intervention of the EGPO algorithm is effective. The policy only needs slight adjustments based on the imperfect teacher to work properly. In other words, the distance between the optimal action and the teacher action is small. However, in more complex environments like Hopper and Walker, the distance between the two is large. As the action-based

intervention is too restrictive, the EGPO algorithm based on such intervention fails to achieve good performance.

In Fig. 6, the performance of EGPO with a SAC policy as the teacher policy is very poor. This is because the employed SAC teacher is less stochastic than the PPO policy. Student's actions have less likelihood in teacher's action distribution and are less tolerated by the action-based intervention function in EGPO, leading to large intervention rate and consequently large distributional shift. Our proposed TS2C algorithm does not access teacher internal action distribution and instead intervenes based on the state-action values of teacher policy, so it is robust to the stochasticity of teacher policy.

