# OpenReview forum: "Guarded Policy Optimization with Imperfect Online Demonstrations"
_ICLR.cc/2023/Conference — ICLR 2023 notable top 25%_

### Official Review · Reviewer_rkHD · 2022-10-23

**Confidence:** 4
**Correctness:** 4
**Technical Novelty And Significance:** 4
**Empirical Novelty And Significance:** 3
**Recommendation:** 8

**Clarity, Quality, Novelty And Reproducibility:**

Well written paper, novel contribution and most details for reproducing the results are shared.

**Strength And Weaknesses:**

Strengths:
- The paper addresses a key problem of learning from an imperfect teacher policy
- It provides performance bounds and thorough empirical evidence

Weaknesses:
- The results are shown only on one environment
- Would be nice to have more details and experiments on how to choose epsilon and if it has to be manually chosen every time based on the environment?

Typos:
- There is a '5' at the beginning of the introduction section


**Summary Of The Paper:**

The paper proposes a novel algorithm to learn from imperfect teacher's policy and achieves super-teacher performance. It provides theoretical justification, performance bounds and experimental results with comparison against strong baselines.

**Summary Of The Review:**

I recommend to accept the paper

---

> ### Author Response · Authors · 2022-11-13
> **Author Response for Reviewer rkHD**
>
> Thank you for your constructive comments. We provide discussions and explanations about your concerns as follows.
>
> Q: **The results are shown only on one environment.**
>
> A: We are conducting additional experiments in other environments. The results will be posted as soon as we finish the experiments.
>
>
> Q: **It would be nice to have more details and experiments on how to choose epsilon and if it has to be manually chosen every time based on the environment?**
>
> A: Epsilon is a hyperparameter in our experiments. Empirically larger epsilon will lead to a more strict intervention criterion and steps with teacher control will be fewer. In order to control the policy distribution discrepancy, we choose epsilon to ensure the average intervention rate to be less than 5%. Nevertheless, different epsilon in the intervention function has little influence on the algorithm performance, as shown in Fig. 6 of our paper. We will add the discussion in the revision.
>
> Q: **There is a '5' at the beginning of the introduction section.**
>
> A: Thanks for pointing it out. We corrected this blunder in the revision.

---

> ### Author Response · Authors · 2022-11-18
> **Updates on Experiment Results**
>
> We conduct an additional experiment in the MuJoCo simulator. The original code is adapted to fit the Hopper environment. We train the PPO algorithm and use its checkpoint with average return 1531 as the teacher policy. When training the TS2C algorithm in the Hopper environment, we follow the warmup-training procedure and keep the hyperparameters unchanged. The comparative results are shown in the following table. Please check Appendix D.2 in the revised paper for the detailed learning curve.
>
>
> | Algorithm                 | TS2C      | SAC       |   EGPO   |
> | ------------------------- |--------|--------- | ----- |
> | Performance at 400k steps | **2735**$\pm$164 | 2040$\pm$256 | 808$\pm$392 |
> | Teacher performance       | 1531      | NA        |   1531   |
>
> As shown in the results, the TS2C algorithm can still achieve best performance in the Hopper environment among baseline algorithms. The SAC algorithm has an unstable performance curve and inferior performance when it converges. The EGPO algorithm shows a constrained performance and can not outperform the imperfect teacher.

---

### Official Review · Reviewer_wVZY · 2022-10-27

**Confidence:** 3
**Correctness:** 4
**Technical Novelty And Significance:** 2
**Empirical Novelty And Significance:** 3
**Recommendation:** 6

**Clarity, Quality, Novelty And Reproducibility:**

* The clarity of quality of the paper is good.
* The novelty is greatly compromised given quite similar previous works that are not discussed in the paper.
* Reproducibility is good since the paper gives detailed experiment setups and the submission includes codes for the experiments.

**Strength And Weaknesses:**

# Strength
* The paper is easy to follow, and the proposed algorithm is simple.
* The experiment results indeed confirm that the student policy could eventually outperform the reference policy, validating the intuition of the algorithm.

# Weakness
* The paper should consider a more diverse set of benchmarks.
* The theory results in the paper are not very illuminating: for example, I am having a hard time understanding the significance of Thm. 3.3: $H$ should also depend on $\epsilon$ because $\epsilon$ defines $\pi_b$. Thus what exactly does the term $\sqrt{H-\epsilon}$ mean?
* The clearest issue of the paper is that the proposed algorithm seems like that the proposed algorithm is highly relevant to AggreVaTe [1] and LOLS [2]. The overall algorithm framework is very similar, and I would say the differences seem rather insignificant. I would assume that works such as [1,2] are overlooked in the literature search, so it would be great to compare and argue in the paper why the contribution of TS2C given the existence these previous works.

[1] Ross, Stephane, and J. Andrew Bagnell. "Reinforcement and imitation learning via interactive no-regret learning."

[2] Chang, Kai-Wei, et al. "Learning to search better than your teacher."

**Summary Of The Paper:**

The paper considers the setting where one is provided a reference policy and we assume that we could be able to estimate or access the value function of the reference policy. The paper thus proposes the method called TS2C where they construct a mixture policy of the student policy and the reference policy and proposes a heuristic to decide which policy to use at run time. The experiment results show that the student policy indeed can overperform the reference policy in practice.

**Summary Of The Review:**

The writing and the experimental results of the paper are good, but at the current stage, I am not convinced that the paper actually makes enough technical contribution given similar previous works. I would recommend a rejection.

======================
Increased the score after the rebuttal.

---

> ### Author Response · Authors · 2022-11-13
> **Author Response for Reviewer wVZY (Part I)**
>
> Thank you for your comments. We provide discussions and explanations about your concerns as follows.
>
> Q: **The clearest issue of the paper is that the proposed algorithm seems like that the proposed algorithm is highly relevant to AggreVaTe and LOLS. The overall algorithm framework is very similar, and I would say the differences seem rather insignificant**
>
> A: The AggreVaTe algorithm is a direct extension of DAgger technique of [1]. It uses the expert cost-to-go as an evaluation of a certain exploration action. It is true that AggreVaTe and TS2C both include the shared control between expert and learner. They also both include the use of expert value. However, there are several significant differences between AggreVaTe and TS2C:
>
> 1. Different in the shift of control. In AggreVaTe, the control is shifted between expert and learner according to the time t which is sampled uniformly from {1, 2, …, T}. Before time t the learner takes control and after time t the expert is in charge. This random shift of control happens only once and is agnostic to different expert and learner policies. In contrast, in TS2C an intervention function, which is dependent on both the expert and learner policies, is used to decide such shifts.
>
> 2. Different in the way of learning the expert value function. TS2C learns the expert value function with Q learning. It learns from the data collected with warmup sampling when training begins. In contrast, AggreVaTe use a Monte-Carlo style of Q-value estimation. It simply records the expert return of one trajectory in the replay buffer.
>
> 3. Different in the way of using the estimated expert value.  In TS2C the expert value function is only used to compute the intervention function. Once the data is collected, the learner trains with its own value function and the expert value function is no longer involved. In contrast, in AggreVaTe the estimated expert value is directly used to train a cost-minimization policy.
>
> 4. Different in the basic learning algorithm. AggreVaTe use classification-based supervised learning to train with the collected data, while TS2C use a reinforcement learning approach, namely the SAC algorithm.
>
> 5. Different in treating imperfect demonstrations. AggreVaTe regards the expert cost-to-go as a reliable estimation of the real and optimal cost-to-go. But in our paper, we take imperfect expert demonstration into consideration and make comparisons between the expert and the learner policies.
>
> 6. Different in the main theory. The theory of AggreVaTe focus on the no-regret learning setting. It gives upper bounds of the learned policy to the optimal policy. In contrast, the theory of TS2C focus on the distribution shift and performance comparison between teacher policy and student policy.
>
> The LOLS algorithm and TS2C both consider a reference policy and try to perform better than it. However, the basic setting of LOLS and TS2C is completely different. LOLS is a “learning to search” (L2S) algorithm. It converts the controlling problem into a search problem with specified search space and actions. It belongs to the general setting of structured contextual bandits. In contrast, TS2C is a value-based deep reinforcement learning algorithm. It is built upon the SAC algorithm and modifies SAC’s sampling method.
>
> In general, AggreVaTe and LOLS consider learning with experts in the context of imitation learning and searching. They are not suitable for complex tasks that require deep reinforcement learning algorithms to solve. TS2C is motivated by the drawbacks of several recent deep IL [2,3] and expert-in-the-loop RL [4] algorithms. Its training pipeline is completely different from those of AggreVaTe and LOLS. We think papers like [2,3,4] are more related to our paper. Therefore, we did not include AggreVaTe and LOLS in the related work.
>
> References:
>
> [1] Stéphane Ross, Geoffrey J. Gordon, Drew Bagnell: A Reduction of Imitation Learning and Structured Prediction to No-Regret Online Learning. AISTATS 2011.
>
> [2] Jonathan Spencer, Sanjiban Choudhury, Matt Barnes, Matthew Schmittle, Mung Chiang, Peter Ramadge, Siddhartha Srinivasa: Learning from Interventions: Human-robot interaction as both explicit and implicit feedback. RSS 2020.
>
> [3] Ajay Mandlekar, Danfei Xu, Roberto Martín-Martín, Yuke Zhu, Li Fei-Fei, Silvio Savarese: Human-in-the-Loop Imitation Learning using Remote Teleoperation. CoRR abs/2012.06733 (2020).
>
> [4] Zhenghao Peng, Quanyi Li, Chunxiao Liu, Bolei Zhou: Safe Driving via Expert Guided Policy Optimization. CoRL 2021.

---

> > ### Comment · Reviewer_wVZY · 2022-12-07
> > **Response**
> >
> > I appreciate the author's response.
> >
> > Regarding my question on theory, I think my concern still remains on the confusion of the $\sqrt{H-\epsilon}$ term. The theory and actual algorithm indeed have a gap, so I believe it might be important that each instant-dependent term in the theory result should at least have an intuitive interpretation (it's not about a tighter bound).
> >
> > After reading the author's response on the differences between the proposed algorithm and previous algorithms, I acknowledge that we may have different interpretations of the significance of these differences, but I still believe it might be beneficial to include at least a discussion on these previous works. I also want to point out that in the differentiable function approximation regime, the update on the supervised loss is actually the same as the policy gradient (see [5]).
> >
> > I appreciate the author's update on the experiment results, and I understand the rebuttal period is short, but I think a more complete evaluation and discussion of the result could make the paper stronger.
> >
> > [5] Sun, Wen, et al. "Deeply aggrevated: Differentiable imitation learning for sequential prediction." International conference on machine learning. PMLR, 2017.

---

> > > ### Author Response · Authors · 2022-12-08
> > > **Author Response for Reviewer wVZY**
> > >
> > > Thank you for your response. We provide discussions and explanations about your concerns as follows.
> > >
> > > Q: **About the term $\sqrt{H-\varepsilon}$, the theory and actual algorithm indeed have a gap, so I believe it might be important that each instant-dependent term in the theory result should at least have an intuitive interpretation.**
> > >
> > > A: We provide intuitive interpretations on the influence of $H$ and $\varepsilon$ here. For reference, the action-based intervention function $T_{action}=1$ when $\mathbb{E}_{a \sim \pi_t(\cdot \mid s)}\left[\log \pi_s(a \mid s)\right]<\varepsilon$. According to Thm 3.3 of our paper, a larger $\varepsilon$ leads to smaller discrepancy between the returns of the behavior and teacher policies. This is because $\varepsilon$ is the threshold for the action-based intervention function. If the action likelihood is less than $\varepsilon$, the teacher policy will take over the control. A larger $\varepsilon$ means more teacher intervention, constraining the behavior policy to be closer to the teacher policy, which leads to a smaller discrepancy in their returns.
> > >
> > > The influence of $H$ can be similarly analyzed. A larger $H$ leads to larger return discrepancy. Intuitively, this is because with higher entropy, the teacher policy tends to have a more “averaged” or multi-modal distribution over the action space. So the policy distributions of the student and teacher are more likely to have overlaps, leading to a higher action likelihood. In turn, the intervention criterion is less likely to be satisfied, leading to fewer teacher interventions.
> > >
> > > In general, the intuitive interpretation of Thm. 3.3 indicates that if we would like larger return discrepancy, i.e. larger performance upper bound as well as smaller lower bound, we should use smaller intervention threshold and teacher policy with higher entropy, and vice versa. Thm. 3.3 has a gap with the actual algorithm in that the algorithm uses a value-based intervention function which is based on Thm. 3.4. Nevertheless, the intuitive interpretation may enlighten future work on how to choose a proper teacher policy in teacher-student shared control. We appreciate your advice and will add this to our paper. To be specific, we will add a brief explanation in the main part of our paper and refer readers to the appendix for full discussion.
> > >
> > > Q: **I still believe it might be beneficial to include at least a discussion on these previous works. I also want to point out that in the differentiable function approximation regime, the update on the supervised loss is actually the same as the policy gradient.**
> > >
> > > A: We agree that including the discussion of aforementioned papers in our paper helps to make the related work section more complete, considering that supervised loss is related to the loss of RL. To be specific, we will discuss AggreVaTe [6] and Deeply AggreVaTeD [7] as extensions of the original DAgger algorithm. They use the expert policy to do a Monte-Carlo style of Q-value estimation. The switching between the learner and the expert happens at a uniformly random time in the trajectory, while we consider an explicit intervention function to decide when the expert takes control. We will also discuss LOLS [8] independently. It also proposes the idea to outperform a reference policy, but belongs to the domain of “Learning to Search”. Our paper is the first to propose that the reference policy can be imperfect and the student policy should have the ability to overtake it in the domain of deep RL.
> > >
> > > Q: **I think a more complete evaluation and discussion of the result could make the paper stronger.**
> > >
> > > A: We have included an additional experiment on the Hopper environment in the MuJoCo simulator in Appendix D.2. Specifically, we train the PPO algorithm and use its checkpoint with average return 1531 as the teacher policy. The comparative results are shown in the following table.
> > >
> > > | Algorithm        |        TS2C      |      SAC    |       EGPO |
> > > | ----- | ----- | ----- | ----- |
> > > |Performance at 400k steps   |          2735$\pm$164|  2040$\pm$256  |   808$\pm$392|
> > > |Teacher performance | 1531 | N.A. | 1531 |
> > >
> > > As shown in the results, the TS2C algorithm can still achieve best performance in the Hopper environment among baseline algorithms. We will also add a discussion on the performance comparison of the MuJoCo and MetaDrive simulator in the experiment part. We hope these additional materials can make the empirical results more concrete.
> > >
> > > We can not update the paper at this time period. The modifications regarding to these three questions will be added to the next version of our paper.
> > >
> > >
> > > **References**
> > >
> > > [6] Ross, Stephane, and J. Andrew Bagnell. "Reinforcement and imitation learning via interactive no-regret learning."
> > >
> > > [7] Sun, Wen, et al. "Deeply aggrevated: Differentiable imitation learning for sequential prediction."
> > >
> > > [8] Chang, Kai-Wei, et al. "Learning to search better than your teacher."

---

> > > > ### Comment · Reviewer_wVZY · 2022-12-12
> > > > **Response**
> > > >
> > > > Thank you for the response. I believe my major concern in the literature part is addressed. I am aware of the additional hopper experiment but by a more complete evaluation, I would appreciate it if more tasks in the benchmark are included. Also additional remarks on the interpretation of the theory part are also appreciated. I will increase my scores accordingly.

---

> > > > > ### Author Response · Authors · 2022-12-13
> > > > > **Thank You for Your Response**
> > > > >
> > > > > We appreciate your response and acknowledgements. We will continue to do the experiments and try to include results on more tasks in the next version of our paper.

---

> ### Author Response · Authors · 2022-11-13
> **Author Response for Reviewer wVZY (Part II)**
>
> Q: **The paper should consider a more diverse set of benchmarks.**
>
> A: We are conducting additional experiments in other environments. The results will be posted as soon as we finish the experiments.
>
> Q: ​​**The theory results in the paper are not very illuminating: for example, I am having a hard time understanding the significance of Thm. 3.3. What exactly does the term $\sqrt{H-\varepsilon}$ mean?**
>
> A: The theory part of this paper serves the role of demonstrating the advantages of the proposed value-based intervention function and the disadvantages of current action-based intervention functions.
>
> First, the upper bound in Eq.(4) shows that with traditional action-based intervention function, the quality of data sampled during training will be constrained by the teacher policy. With imperfect teacher demonstrations, the trained policy can have poor performance.
>
> Second, the difference in the order of $1-\gamma$ between Eq.(4) and Eq.(5) shows that the value-based intervention function can be more efficient in injecting teacher knowledge in training (as the student performance lower-bound is closer to the teacher’s performance), compared with action-based intervention functions. The term $\sqrt{H-\varepsilon}$ is included in its current form without further deduction mainly to keep the equation simple and to the point. We did not derive a tighter bound since the derivation will not be related to the main contribution of this paper.  Thm. 3.3 and Thm. 3.4 in their current form are enough to demonstrate that the value-based intervention function has the advantage of providing more efficient exploration and better safety guarantee compared with action-based intervention function.

---

> ### Author Response · Authors · 2022-11-18
> **Updates on Experiment Results**
>
> We conduct an additional experiment in the MuJoCo simulator. The original code is adapted to fit the Hopper environment. We train the PPO algorithm and use its checkpoint with average return 1531 as the teacher policy. When training the TS2C algorithm in the Hopper environment, we follow the warmup-training procedure and keep the hyperparameters unchanged. The comparative results are shown in the following table. Please check Appendix D.2 in the revised paper for the detailed learning curve.
>
>
> | Algorithm                 | TS2C      | SAC       |   EGPO   |
> | ------------------------- |--------|--------- | ----- |
> | Performance at 400k steps | **2735**$\pm$164 | 2040$\pm$256 | 808$\pm$392 |
> | Teacher performance       | 1531      | NA        |   1531   |
>
> As shown in the results, the TS2C algorithm can still achieve best performance in the Hopper environment among baseline algorithms. The SAC algorithm has an unstable performance curve and inferior performance when it converges. The EGPO algorithm shows a constrained performance and can not outperform the imperfect teacher.

---

### Official Review · Reviewer_oviQ · 2022-10-31

**Confidence:** 3
**Correctness:** 3
**Technical Novelty And Significance:** 3
**Empirical Novelty And Significance:** 3
**Recommendation:** 5

**Clarity, Quality, Novelty And Reproducibility:**

The paper is clearly written, with both algorithms and experimental setup quite clearly explained. In terms of contributions, the main novelty is TS2C, which is a flexible TSF framework that combines with off-policy RL student improvement and a value-aware methodology of deciding when a teacher policy should intervene the student.

TS2C can work with a wide range of teacher policies with different performance. Finally the authors evaluate the TS2C algorithm on a wide range of benchmark experiments and detailedly demonstrate the ability of TS2C in an autonomous driving scenario, which shows the algorithms' ability not only on imitation learning (of teacher) but also performance improvement guarantees. Contribution is slightly more on the empirical sense in which a new TSF framework is proposed to solve the problem of imperfect teacher. Algorithmically the novelty is moderately significant and most experimental hyperparameters and setup are explained in great details in the paper and its appendix to help readers better understand the experiment. However, the source code is not available reproducibility.

**Strength And Weaknesses:**

Strengths:
The paper is well-written with concepts clearly explained.
The problems studied in this work is important and the concerns of existing TSF work where the teacher is always optimal is a realistic one with good real-world implications.
Experiment is thorough, validating the student policy can outperform the teacher in the TS2C algorithm. It is also backed with some theoretical analysis to justify this approach

Weaknesses:
Discussions of some experimental parameters are still missing. For example it is still unclear how the epsilon parameter is chosen.
The theory part of this work is very straight-forward, the bound developed is pretty much a direct result from existing work.
especially lemma 1 and theorem 3.2 are pretty much standard results from occupancy distribution matching and policy-value differences in RL (e.g., TRPO-style proofs).
How tight is the bound in Theorem 3.3 (it seems with the (1-\gamma)^2 in the denominator the bound is rather loose), and how to effectively determine the average entropy in practice to validate the tightness of this bound?


**Summary Of The Paper:**

In this work, the authors proposed a teacher-student framework (TSF), where a
teacher agent or human expert guards the training of a student agent by intervening and providing online demonstrations that the teacher policy may not be optimal. This is a more realistic setting than standard TSF in which the teacher policy is always assumed to be near-optimal. Specifically, they develop a new method that can incorporate arbitrary teacher policies with
modest performance and utilize an off-policy RL method known as Teacher-Student Shared Control (TS2C) to effectively update the student policy. In this method the main novelty is to incorporate teacher intervention based on trajectory-based value estimation. Theoretical analysis justifies also that the proposed algorithm attains efficient exploration and lower-bound safety guarantee without being affected by the teacher’s
own performance, and this algorithmic finding is also furthered validated with several benchmark experimentations (especially with autonomous driving simulation), showcasing that TS2C can exploit teacher policies at any performance level, maintain lower
training cost and the corresponding student policy can often outperform the imperfect teacher policy.

**Summary Of The Review:**

The work is quite neat in terms of addressing the teacher-student framework in RL problem in which the teacher does not necessarily need to be near-optimal. The TS2C framework is also quite flexible and can be adapted to different off-policy RL algorithms. Experiments show the superiority of this algorithm in terms of imitating the teacher, deciding when the teacher should intervene the student during training and having the learned student. outperforming the teacher.

Novelty wise I think the underlying technique has merits in terms of problem formulation, algorithms and experimentation, albeit the theoretical analysis being a bit straight-forward, which is my main concern of the technicality of this work.

---

> ### Author Response · Authors · 2022-11-13
> **Author Response for Reviewer oviQ**
>
> Thank you for your constructive comments. We provide discussions and explanations about your concerns as follows.
>
> Q: **Discussions of some experimental parameters are still missing. For example it is still unclear how the epsilon parameter is chosen.**
>
> A: Epsilon is a hyperparameter in our experiments. Empirically larger epsilon will lead to a more strict intervention criterion and steps with teacher control will be fewer. In order to control the policy distribution discrepancy, we choose epsilon to ensure the average intervention rate to be less than 5%. Nevertheless, different epsilon in the intervention function has little influence on the algorithm performance, as shown in Fig. 6 of our paper. The coefficient for intervention minimization $\lambda$ is simply set to 1. If used in other environments, it may need some adjustments to fit the reward scale. The coefficient for maximum entropy learning $\alpha$ is updated during training as in the SAC algorithm. The number of warmup timesteps is empirically chosen so that the expert value function can be properly trained. Other parameters follow the setting in EGPO [1]. For example, hidden layer sizes are set to (256, 256) and the activation function is RELU.  We will add the description and discussion in the revision.
>
>
> Q: **The theory part of this work is very straight-forward, the bound developed is pretty much a direct result from existing work.**
>
> A: The theory is not the main contribution of this work. It serves the role of demonstrating the advantages of the proposed value-based intervention function and the disadvantages of current action-based intervention functions. First, the upper bound in Eq.(4) shows that with traditional action-based intervention function, the quality of data sampled during training will be constrained by the teacher policy. With imperfect teacher demonstrations, the trained policy can have poor performance. Second, the difference in the order of $1-\gamma$ between Eq.(4) and Eq.(5) shows that the value-based intervention function can be more efficient in injecting teacher knowledge in training (as the student performance lower-bound is closer to the teacher’s performance), compared with action-based intervention functions. Apart from that, Lemma 3.1 and Thm. 3.2 first introduces discrepancy-based analysis to the setting of teacher-student shared control. The analysis provides a new explanation on the advantage of shared control over imitation learning.
>
>
> Q: **How tight is the bound in Theorem 3.3, and how to effectively determine the average entropy in practice to validate the tightness of this bound?**
>
> A: Many previous works (Thm. 1 in [2], Thm. 4.1 in [3], Thm. 1 in [4])  have shown that $(1-\gamma)^2$ will appear in the denominator when it comes to differences of the cumulative return, given the difference in the action distribution. Intuitively, aggregating the difference of $\varepsilon$ in the action distribution across the trajectory will lead to a difference of $\frac{C\varepsilon}{1-\gamma}$ in the value function. The greedy policy derived from the value function will in turn have a difference of $\frac{C\varepsilon}{1-\gamma}$ in the reward at each step. Aggregating the difference across the trajectory will lead to a difference of $\frac{C\varepsilon}{(1-\gamma)^2}$ in accumulated return. The $(1-\gamma)^2$ term in Thm. 3.3 is tight, which dominates the gap in accumulated return. Nevertheless, the other constant terms, e.g. Rmax and the average entropy, can be tighter given some additional assumptions. We did not derive a tighter bound since the derivation will not be related to the main contribution of this paper, which is the new type of intervention function. Thm. 3.3 and Thm. 3.4 in their current form are enough to demonstrate that the value-based intervention function has the advantage of providing more efficient exploration and better safety guarantee compared with action-based intervention function. We will add this discussion in the revision.
>
>
> Q: **The source code is not available for reproducibility.**
>
> A: The source code has been included in the supplementary materials. Please check the supplementary/code directory and the README.md file for steps to reproduce the results.
>
> References
>
> [1] Zhenghao Peng, Quanyi Li, Chunxiao Liu, Bolei Zhou: Safe Driving via Expert Guided Policy Optimization. CoRL 2021.
>
> [2] Tian Xu, Ziniu Li, Yang Yu: Error Bounds of Imitating Policies and Environments. NeurIPS 2020.
>
> [3] Michael Janner, Justin Fu, Marvin Zhang, Sergey Levine: When to Trust Your Model: Model-Based Policy Optimization. NeurIPS 2019.
>
> [4] John Schulman, Sergey Levine, Pieter Abbeel, Michael I. Jordan, Philipp Moritz: Trust Region Policy Optimization. ICML 2015.

---

### Official Review · Reviewer_L3ge · 2022-11-04

**Confidence:** 4
**Correctness:** 4
**Technical Novelty And Significance:** 3
**Empirical Novelty And Significance:** 3
**Recommendation:** 8

**Clarity, Quality, Novelty And Reproducibility:**

The paper is well written and clear. The core concept of agent switches here is not new, but the particular implementation allows the student to learn and eventually outperform a sub-optimal teacher, unlike baseline methods.

**Strength And Weaknesses:**

Strengths:

1. The proposed algorithm is intuitive and straight-forward
2. It asymptotically outperforms baselines
3. The proposed method is backed by theoretical analysis

Weaknesses

1. It seems that the algorithm takes some time before it starts improving and is initially substantially outperformed by EGPO. Is this because at the start of the training the switch allows most student actions, while EGPO mostly uses the teacher?

Question:

What is the effect of the optimization algorithm here? The proposed method uses Q-ensembles, which combined with more often training have demonstrated significant improvement in sample complexity (RedQ). What update frequency was used in this paper and is this directly comparable to the baseline methods?


**Summary Of The Paper:**

The paper studies the Teacher-Student Framework, in the case when the teacher is potentially sub-optimal. Learning here is based on an ensemble off-policy method. The core concept is to develop a teach intervention function that is based on the estimated sub-optimality of student actions with respect to the teacher's value function. The advantage of the proposed method is that it allows for continuous student improvement, to the point that the student takes complete control in the case of sub-optimal teacher. The authors provide theoretical bounds for the proposed method and evaluate it on a driving simulator. The proposed method 1) outperforms baselines across evaluation scenarios and 2) demonstrates the student capability to solve the task, even with a significantly sub-optimal teacher.

**Summary Of The Review:**

Simple and intuitive method to allow student agent to learn and outperform a potentially sub-optimal teacher. Clear writing with good theoretical backing.

---

> ### Author Response · Authors · 2022-11-13
> **Author Response for Reviewer L3ge**
>
> Thank you for your constructive comments. We provide discussions and explanations about your concerns as follows.
>
> Q: **It seems that the algorithm takes some time before it starts improving and is initially substantially outperformed by EGPO.**
>
> A: This is because in the TS2C algorithm the intervention function requires an estimator for the teacher policy’s Q-value function. We allocate 50k timesteps of environment interaction to allow the teacher policy to collect data in the environment and train a teacher Q-network. These data will not be used to train the student policy. As such, the test reward of the student policy trained by TS2C remains almost unchanged for the first 50k timesteps of training. The baseline algorithm Importance Advising also requires a Q-value estimator, so its training curve shows similar patterns.
>
> Q: **What is the effect of the optimization algorithm here? What update frequency was used in this paper and is this directly comparable to the baseline methods?**
>
> A: In our paper, the ensemble of Q-networks is only used in the intervention function. In computing the target Q-value for optimization, we only use the first two Q-networks, as in the SAC algorithm. Therefore, the UTD (Update-to-Data) ratio is still set to 1, which is the same across our method and all of the baseline methods. This helps to make a fair comparative analysis. Nevertheless, the attempt of using all ensemble Q-networks as in [1] and increasing the UTD ratio is compatible with our TS2C algorithm.
>
> References
>
> [1] Xinyue Chen, Che Wang, Zijian Zhou, Keith W. Ross: Randomized Ensembled Double Q-Learning: Learning Fast Without a Model. ICLR 2021.

---

### Author Response · Authors · 2022-11-13
**Changelog of Paper Revision**

[2022-11-13] The discussions on the theoretical results are added in Appendix A.2. The details of choosing hyper-parameters are added in Appendix C.3. We also fix some typos in the revision.

[2022-11-18] The experiment results on the Hopper environment is added in Appendix D.2.

---

### Decision · Program_Chairs · 2023-01-20

**Decision:**

Accept: notable-top-25%

**Justification For Why Not Higher Score:**

I'm trying to gauge the potential impact based on reviewer feedback, and would be open-minded for the score to be higher (i.e., oral).

**Justification For Why Not Lower Score:**

This paper is rigorous and innovative.  I think the community would be interested in this contribution.

**Metareview: Summary, Strengths And Weaknesses:**

This work introduces a version of Teacher-Student online learning that allows for student actions to dominate when there is anticipated high value, even if they violate the behavior acceptable to the teacher.  The estimate of how valuable student actions may be comes from a learned Q function.

The reviews for this paper were mixed, with two reviewers finding the proposed approach to be an intuitive solution to an important problem and to be theoretically well justified.  Reviewer oviQ, who gave a score of a 5, found the paper to be well written and tackle an important problem, but found the theoretical analysis to be too straightforward.  The authors replied that the theoretical analysis is merely an assurance and not a primary contribution.  I think this is a reasonable response.  Reviewer wVZY rated the paper a 3, and their review points out that the paper is clear, the algorithm is simple, and the experiments support the theory.  However, this reviewer believed that more diverse benchmarks were required and there is some relevant literature that is not compared that possibly is very similar.

Overall, in my opinion, there is some fundamental novelty to this approach, and the work is quite rigorously conducted.  However, there are a few high level points where I think things are slightly imprecise and I would encourage the authors to make a few additional changes before the camera-ready final version:
- The first sentence of the abstract asserts that the Teacher-Student Framework involves interventions that override student actions that are deemed unsafe.  As pointed out in the first paragraph of the main text, I'm not sure that interventions for safety is always assumed in Teacher-Student settings.  This is to say, there is a little more slack in how this framework is used and what it assumes than you imply in the abstract (e.g. a teacher inducing exploration is not "guarding" the student behavior).  I would encourage the authors to basically acknowledge that the version they compare against is one important teacher-student setting rather than the somehow primary or canonical one.
- Also, for example, when the authors state "This idea resembles imitation learning (IL) methods", they actually mean more precisely "online imitation learning" approaches.  As a related note, while I do think the present work is distinct from existing online imitation learning approaches, I share Reviewer wVZY's concern that the work does not appropriately discuss the connections to some of those previous papers.
- As a conceptual point, I'm not totally satisfied with there being a Q function from the experts that is always used to determine whether the student retains control versus the expert intervening.  As the Q function associated with the student learning develops, couldn't that be leveraged?  Basically, it just seems like there must be a large fraction of states for which the expert Q functions are non-informative (i.e. the ensemble has high variance) but where the student Q function has developed confidence.  In these regions, it seems like the expert will retain control, even though it perhaps should not. It would be worth discussing this design choice somewhere as it seems potentially a bit ad hoc.

**Note From Pc:**

if the above contains the word "oral" or "spotlight" please see: "oral" presentation means -> notable-top-5% and "spotlight" means -> notable-top-25%. As stated in our emails, we are disassociating presentation type from AC recommendations

**Summary Of Ac-Reviewer Meeting:**

Due to the conflicting scores, we met via a video chat to discuss the paper.  Unfortunately reviewer oviQ was not able to participate.  Reviewer L3ge reiterated their support for the paper and emphasized that this setting is important and understudied insofar as the teacher is suboptimal.  Reviewer wVZY emphasized that the major remaining concern was that there is previous work that is similar, in particular pointing to AggreVaTe (as mentioned in their review).  The two other reviewers present challenged this, not seeing AggreVaTe as especially relevant as a baseline.  A consensus was reached that while AggreVaTe is not relevant enough to be a direct comparison, it could be cited/discussed in the paper.  Reviewer rkHD found the work sufficiently novel and highlighted the innovative use of the value estimate to decide whether to use the teacher.  Reviewer wVZY stated that if the authors added more evaluation, clarified the theory point in their review (they were not satisfied with the author reply), and clarified the connection to previous work (the authors didn't indicate they would add discussion of the mentioned references), then the paper score would have been updated to a 6-8.  I encouraged reviewer wVZY to state this for the authors in a reply to their comments; the reviewer and author had a final exchange, after which the reviewer increased their score to a 6.

Given when this feedback was posted, the authors can no longer revise their paper, but they have committed to making the remaining changes.